

# Alluvial record of an early Eocene hyperthermal, Castissent Formation, Pyrenees, Spain

Louis Honegger[1], Thierry Adatte[2], Jorge E. Spangenberg[3], Jeremy K. Caves Rugenstein[4], Miquel Poyatos-Moré[5], Cai Puigdefabregas[6], Emmanuelle Chanvry[7], Julian Clark[8], Andrea Fildani[8], Eric Verrechia[2], Kalin Kouzmanov[1], Matthieu Harlaux[1], Sébastien Castelltort[1]

[1]Department of Earth Sciences, University of Geneva, Rue des Maraîchers 13, 1205 Geneva, Switzerland
[2]Institut of Earth Sciences, Géopolis, University of Lausanne, 1015 Lausanne, Switzerland
[3]Institute of Earth Surface Dynamics (IDYST), Géopolis, University of Lausanne, 1015 Lausanne, Switzerland
[4]Geological Institute, ETH Zürich, Sonneggstrasse 5 NO E61, 8092 Zürich, Switzerland
[5]Department of Geosciences, University of Oslo, Sem Sælands vei 1, 0371 Oslo, Norway
[6]Department of Earth and Ocean Dynamics, University of Barcelona, C/ Martí i Franquès, s/n, 08028 Barcelona, Spain
[7]University of Poitiers & CNRS, IC2MP, INC, 86000 Poitiers, France
[8]Equinor Research Center, 6300 Bridge Point Parkway, Building 2, Suite 100, Austin, Texas, USA

*Correspondence to*: Louis Honegger (*louis.honegger@unige.ch*)

**Abstract.** During the late Palaeocene to the middle Eocene (57.5 to 46.5 Ma) a total of 39 hyperthermals—periods of rapid global warming recorded by prominent negative carbon isotope excursions (NCIEs) as well as peaks in iron content—have been recognized in marine cores. Understanding how the Earth system responded to rapid warming during these hyperthermals is fundamental because they represent potential analogues, in the geological record, to the ongoing anthropogenic modification of global climate. However, while hyperthermals have been well documented in the marine sedimentary record, only few have been recognized and described in continental deposits, thereby limiting our ability to understand the effect and record of global warming on terrestrial surficial systems. Hyperthermals in the continental record could be a powerful correlation tool to help connect marine and continental records, addressing issues of environmental signal propagation from land to sea. In this study, we generate new stable carbon isotope data ($\delta^{13}$C values) across the well-exposed and time-constrained fluvial sedimentary succession of the early Eocene Castissent Formation in the South-Central Pyrenees (Spain). The $\delta^{13}$C values of pedogenic carbonate reveal—similarly to the global records— stepped NCIEs, culminating in a minimum $\delta^{13}$C value that we correlate with the hyperthermal event "U" at *ca.* 50 Ma. This general trend towards more negative values is most probably linked to higher primary productivity leading to an overall higher respiration of soil organic matter during these climatic events. The relative enrichment in immobile elements (Zr, Ti, Al) and higher estimates of mean annual precipitation together with the occurrence of small iron-oxides/hydroxides nodules during the NCIEs suggest intensification of chemical weathering and/or longer exposure of soils in a highly seasonal climate. The results show that even relatively small-scale hyperthermals compared with their prominent counterparts, such as PETM, ETM2 and 3, have left a recognizable trace in the stratigraphic record, providing insights into the dynamics of the carbon cycle in continental environments during these events.



# 1 Introduction

Starting at the end of the Palaeocene, a period of global warming reached its climax during the Early Eocene Climatic Optimum (EECO) (Hyland and Sheldon, 2013; Westerhold and Röhl, 2009). The EECO started *ca.* 53 Ma ago and lasted until *ca.* 49 Ma ago (Westerhold et al., 2018), after which the climate began to cool towards icehouse conditions eventually reached later in the Cenozoic (~Eocene-Oligocene transition, Zachos et al., 2001, 2008). Superimposed on, and coeval to, this globally warm epoch, brief-periods of pronounced global warming known as "hyperthermals" standout as anomalies outside of background climate variability (Dunkley Jones et al., 2018; Turner et al., 2014). The Palaeocene-Eocene Thermal Maximum (PETM; ~56 Ma) was the first of these events to be identified globally because of its exceptional magnitude and preservation in both marine and continental deposits (Koch et al., 1992). To date, for the late Paleocene – early Eocene period, a total of 39 hyperthermal events of lesser magnitude have been identified from marine cores (e.g., Westerhold et al., 2018), among which the most prominent and studied are the Early Thermal Maximum (ETM) 2, H2, I1, I2, and ETM3/X events (Deconto et al., 2012; Lourens et al., 2005; Lunt et al., 2011; Nicolo et al., 2007; Turner et al., 2014) (Fig. 1). In the stratigraphic record, these events are primarily characterized by important negative carbon isotope excursions (NCIEs) exceeding background variability (Abels et al., 2016; Cramer et al., 2003; Lauretano et al., 2016; Nicolo et al., 2007; Sluijs and Dickens, 2012; Zachos et al., 2008), i.e. typically with amplitude greater than the standard deviation (SD) of pre-hyperthermal background values.

In deep marine settings, the NCIEs are typically paired with an increase in iron concentration and decrease in carbonate content, indicating ocean acidification potentially linked with high atmospheric $CO_2$ concentrations (Nicolo et al., 2007; Slotnick et al., 2012; Westerhold et al., 2018). In coastal marine sections, Early Eocene hyperthermal events are generally associated with an enhanced flux of terrigenous material, interpreted as linked to accelerated hydrological cycle and higher seasonality (Bowen et al., 2004; Dunkley Jones et al., 2018; Nicolo et al., 2007; Payros et al., 2015; Slotnick et al., 2012), although several studies document a spatially heterogeneous hydrological climatic response during the PETM (Bolle and Adatte, 2001; Carmichael et al., 2017; Kraus and Riggins, 2007). In fluvial systems, the abrupt warming of the PETM was found to be associated with expansion and coarsening of alluvial facies combined with an increase of the magnitude of flood discharge (Chen et al., 2018; Foreman et al., 2012; Pujalte et al., 2015), as well as enhanced pedogenesis (Abels et al., 2012). Yet, how continental systems reacted to the other, smaller-magnitude hyperthermals of the Early Eocene remains to be documented. In particular, because of the subaerial nature and important lateral dynamics of alluvial systems (e.g., Foreman and Straub, 2017; Straub and Foreman, 2018), the extent to which fluvial successions can provide complete and faithful archives of past climatic events, especially those with the smallest magnitudes, is a matter of debate (Foreman and Straub, 2017; Straub and Foreman, 2018; Trampush et al., 2017). Addressing this question is particularly critical for studies attempting to understand environmental signal propagation in source-to-sink system (e.g., Castelltort and Van Den Driessche, 2003; Duller et al., 2019; Romans et al., 2016; Schlunegger and Castelltort, 2016), which require high-resolution continental-marine correlations such as those provided by the PETM (e.g., Duller et al., 2019) but also by other hyperthermals of the Early Eocene.

To address these issues, we explored some aspects of the geochemical signature (carbon and oxygen stable isotopes, major and trace elements) and of the sedimentology of the fluvial Castissent Fm (South-Central Pyrenees, Spain, Fig. 2), whose deposition took place during the late EECO. First, we generated a new carbon





isotope profile from a paleosol succession rich in carbonate nodules across the Castissent Fm in order to compare these results with a global $\delta^{13}$C record. The data suggest that this fluvial succession preserves a record of hyperthermal "U" event at *ca.* 50 Ma, adding important constraints to the age of this Formation. Second, we used the major and trace element composition of bulk floodplain material in order to explore the climatic impact of such minor hyperthermal, including empirical reconstructions of mean annual precipitation, allowing us to discuss soil dynamics during global warming. This study identifies for the first time in a continental succession an event so far only recorded in marine sediments, thereby demonstrating the global breadth of these climatic events and the complementarity of oceanographic and terrestrial archives.

## 2 Geological setting

The Castissent Formation is a Ypresian age fluvial Formation that crops out in the Tremp-Graus Basin (South-Pyrenean foreland basin, Marzo et al., 1988, Fig. 2), which developed during the Paleocene to Eocene and is bounded by the Bóixols thrust in the North, and the Montsec thrust in the South (Marzo et al., 1988). The Castissent Fm is defined by its prominent overall sand-rich character, and is composed in detail of three coarse-grained multistorey channels complexes (labelled as Members A, B and C) separated by four marine incursions (M0 to M3) inferred from the observation of marginal coastal bioclast-rich horizons developed up into the upper deltaic plain and correlative with finer dark-grey mudstones and calcretes in the fluvial segment of the Castissent (Marzo et al., 1988). This major fluvial progradation is correlated westwards with deep-water turbidite sequences of the Arro and Fosado Formations in the Ainsa Basin (Fig. 3, Mutti et al., 1988; Nijman and Nio, 1975; Nijman and Puigdefabregas, 1978; Pickering and Bayliss, 2009). In the upstream, eastern counterparts of the Castissent Fm, the channel complexes are intercalated with yellow to red coloured paleosols. Sub-spherical to slightly elongated carbonate nodules with a diameter ranging from 1 mm to 4 cm are omnipresent in the paleosols. Studies of the Castissent Fm tentatively attributed its occurrence to an important pulse of exhumation and thrust activity in the hinterland at *ca.* 50 Ma, in possible combination with a late-Ypresian sea-level fall (Castelltort et al., 2017; Marzo et al., 1988; Puigdefabregas et al., 1986; Whitchurch et al., 2011), both resulting in reduced available accommodation space enhancing progradation and amalgamation (Chanvry et al., 2018).

Constraints on the age of the Castissent Fm in its upstream segment is the recognition of European Mammals zone MP10 (Checa-Soler, 2004; Payros and Tosquella, 2009), which gives a broad age range of between 50.73 to 47.4 Ma (GTS2012). However, most age constraints have been obtained through bio- and magnetostratigraphic studies in the downstream more marine-influenced segment of the Castissent Fm in the Campo area located 40km westward from the Chiriveta section (fig. 3). Thanks to its very prominent field expression, the Castissent Fm has been physically mapped from west to east across these sections (Nijman, 1998; Nijman and Nio, 1975; Poyatos-Moré, 2014) and the stratigraphic constraints obtained in the west can thus be propagated eastwards to its more fluvial counterparts (Fig. 3). Kapellos and Schaub (1973) find the transition between the *D. lodoensis* and the *T. orthostylus* nannoplankton (NP) zones at *ca* 200 m below the base of the Castissent Fm and the transition between the *T. orthostylus* and the *D. sublodoensis* NP zones in the transgression *ca* 100 m above the uppermost Member of the Castissent Fm. This indicates that the Castsissent Fm was deposited during NP13. Magnetostratigraphic data of the same section by Bentham and Burbank (1996) place the transition between C22r and C22n magnetozones closely above the top of the Castissent Fm. We thus used the recent astrochronologic age models of Westerhold et al. (2017), which obtain numerical ages of



50.777±0.01 and 49.695±0.043 Ma respectively for the base and top of C22r, and obtain a numerical age of 50.534±0.025 Ma for the base of NP13 based on ODP site 1263. We considered that these constraints give a maximal age extension of between 50.534±0.025 and 49.695±0.043 Ma for the Castissent Fm (reported in green on fig. 1). According to global isotopic records (Fig. 1), this period was marked by 4 hyperthermals labelled S, T, U and V (Lauretano et al., 2016; Westerhold et al., 2017).

## 3 Material and methods

The Chiriveta section is situated in a continental paleogeographic position prone to pedogenesis and slightly off-axis from the more "in-axis" amalgamated sand-rich type section of Mas de Faro (Fig. 2); for paleo-position and correlation see also Figs. 10 and 12 in Marzo et al. (1988).

### 3.1 Sampling

A total of 74 samples were collected from the early-Eocene Chiriveta section for geochemical studies. All samples consist of floodplain material and were taken below the weathering depth (~50 cm), with an average resolution of 1 m. Resolution was increased by a factor of 2 in specific horizons such as red beds. When important sandbodies occurred, lateral equivalent floodplain material or intercalated paleosol horizons were sampled. Each sample was split in two aliquots, one for major and trace element analysis and the other for carbon and oxygen stable isotope analysis on pedogenic carbonate nodules. The carbonate nodules were extracted from the bulk paleosol material by sieving and then cleaned by repeated washes with deionized water in an ultrasound bath. From each cleaned nodules set, subsamples of 1 to 4 nodules were taken, leading to a total of 149 sub-samples of pedogenic carbonate nodules.

### 3.2 Carbon and oxygen stable isotopes

Pedogenic carbonate nodules were crushed and powdered in an agate mortar and analysed for stable carbon and oxygen isotope composition at the Institute of Earth Surface Dynamics of the University of Lausanne (Switzerland) using a Thermo Fisher Scientific (Bremen, Germany) carbonate-preparation device and Gas Bench II connected to a Thermo Fisher Delta Plus XL isotope ratio mass spectrometer. The carbon and oxygen isotope compositions are reported in the delta ($\delta$) notation as the per mil (‰) isotope ratio variations relative to the Vienna Pee Dee Belemnite standard (VPDB). The analytical reproducibility estimated from replicate analyses of the international calcite standard NBS-19 and the laboratory standard Carrara Marble was better than ± 0.05 ‰ (1 sigma) for $\delta^{13}$C and ± 0.1 ‰ (1 sigma) for $\delta^{18}$O.

### 3.3 Major and trace element composition

Fifty-two bulk paleosol samples were analysed for major and trace elements using X-Ray fluorescence (XRF) 150   spectrometry. Crushed bulk powders (<80 μm) were mounted in a plastic cup covered by a thin polypropylene film (4 μm thick) and analysed in the laboratory with a Thermo Niton XL3t® portable XRF analyzer fixed on a test stand. Analyses were performed with a beam diameter of 8 mm, to determine the concentrations of 34 major and trace elements (from Mg to Au). Each measurement took 120 s, consisting of two 60 s cycles on four different filters (15 seconds on low, main, high, and light ranges), operating the X-ray tube at different voltage to



optimize the fluorescence and peak/background ratios of the different elements. The limits of detection were of
10's ppm for most elements, except for Mg, Si, and Al which are at wt% level. Na is too light to be detected. The
spectra of the measurements were transferred to a computer using NDT software version 8.2.1. (Thermo Fisher
Scientific, Waltham, Ma, USA). Twenty-three major and trace elements were analysed on pressed and fused
discs of the same material using a PANalytical PW2400 XRF spectrometer at the University of Lausanne to
cross-calibrate the compositions measured with the Niton XL3t® portable XRF analyzer.

### 3.4 Mean annual precipitation

The mean annual precipitation estimate (MAP) used in this study was estimated from the empirical relationship
between MAP and $CaO/Al_2O_3$ ratio for Mollisols from a national survey of North American soils according to
the following equation: MAP (mm) = $-130.9*ln(CaO/Al_2O_3) + 467$ (Sheldon et al., 2002). CaO and $Al_2O_3$
concentrations were measured on bulk paleosol material. Climate linked to the MAP estimate was classified
based on the following boundaries: arid to semiarid at 250 mm and semiarid to subhumid at 500 mm (Bull,
1991).

### 3.5 Grain-size estimation

The relative grain-size variation of the sediment samples was estimated from their major element compositions.
Si, Ti and Zr are more concentrated in the coarse fraction of the sediment as they are found in larger mineral
grains, whereas Al is more concentrated in the finer fraction of the sediment because is mostly linked to clay
minerals (Croudace and Rothwell, 2015; Lupker et al., 2011, 2012). Grain size variation throughout the section
was estimated using Si/Al, Ti/Al and Zr/Al ratios, therefore, an increase in these ratios suggests a relative
increase in the proportion of coarser material in the sample.

### 3.6 Correlation with target curves

The measured $\delta^{13}C$ dataset was compared with a time-equivalent ODP 1263 global $\delta^{13}C$ record reported by
Westerhold et al. (2017) using the Analyseries software (Paillard et al., 1996). The $\delta^{13}C$ record of site 1263 was
favoured over those of ODP 1209 and 1258 covering the Castissent Fm time-period, because it is continuous and
has a higher resolution. Correlations between the $\delta^{13}C$ record of site 1263 and the $\delta^{13}C$ record of the Chiriveta
section were performed in order to optimize the Pearson correlation coefficient ($r$) and by minimizing abrupt
variations in sedimentation rates. Well-defined peaks in both $\delta^{13}C$ records were used as tie-points for the
correlation and the number of tie-points was kept minimum (<10) so as not to force the correlations.

### 4 Results

#### 4.1 Overview of the Castissent Fm at the Chiriveta section

In the proximal part of the Basin, the Castissent Fm is a paleosol rich succession, which shows greyish-yellow to
red-brown mottled floodplain paleosols (Fig. 4A-B-D-E), which correspond laterally to thick, medium to coarse-
grained quartz-rich channel-fill deposits (width/depth ratio = 20-50; Marzo et al. (1988)) and over-bank deposits
with an overall westerly flow direction parallel to the main structures of the growing Pyrenean orogeny (Marzo
et al., 1988). At the base of the section, the first marine incursion M0 is situated at the top of a 20 m-thick





coarse-grained tidal bar deposit with herringbone cross-stratifications and oyster shells (Fig. 4C). In the
        Chiriveta section, the Castissent Member A is a 48 m-thick interval comprising two main white medium-grained
        sandbodies of 5.40 and 1.5 m in thickness respectively. Sandy bed-forms observed in the first sandbody have a
        mean height of 24 cm ($n$ = 9). The second marine incursion M1 is located at 48 m just below the Castissent B
        Member and consists of a 2 m-thick grey interval interpreted by Marzo et al. (1988) as poorly drained brackish
water facies (Fig. 4B-F). The Castissent B Member (Fig. 4G) begins with a 12 m-thick and laterally-extensive
        (width/depth ratio ≥ 250; Marzo et al. (1988)) amalgamated sandbody with a micro-conglomeratic erosive base.
        Grain size is overall larger than in Member A, and ranges from fine sand to large pebbles. Sandbodies tops show
        a fining-upward trend and are capped by mottled siltstone packages. Mottled siltstone layers are interpreted as
        pedogenized over-bank deposits based on roots traces and their capping relationship with underlying sandbody
deposits (observed at 26 m, 76 m, 89 m and 96 m in Fig. 4, Fig. 4H). More regular and sheet-like sandbodies
        interbedded with several mottled siltstone layers are observed upwards. The section ends with a 23 m-thick,
        medium to very coarse tidally-influenced sandstone deposits interpreted as the equivalent M3 marine incursion
        by Marzo et al. (1988). Although Castissent Member C was not interpreted by Marzo et al. (1988) in this section,
        a 2m-thick fine-grained sandbody at ca 80 meters on our section may be a condensed lateral equivalent of it (Fig.
205     5).

### 4.2 Stable isotopic record

        Carbon and oxygen isotope ratios from the carbonate nodules are presented in Fig. 5. The $\delta^{13}$C values vary
        between -10.9 and -1.9‰ with a mean value and 1 SD of $-7.7 \pm 1.6$ ‰. Six NCIE (named A to F in Fig. 5 and
        colour coded in Fig. 6) are more negative than the mean value − 1 SD amongst which one (NCIE D) is below 2
SDs. The values are $-9.6$, $-9.8$, $-9.9$, $-10.9$, $-9.9$ and $-9.4$‰ for NCIEs A to F respectively. At the bottom of the
        section, NCIE A is followed by a relatively constant interval of mean $\delta^{13}$C values. NCIE B, situated in the first
        red bed, marks the beginning of a stepped $\delta^{13}$C trend (around ±1 SD) leading to the minimum NCIE D. The
        second part of the section shows two more NCIE separated by the highest $\delta^{13}$C value at 65 m. NCIE F is the
        lowest of all NCIEs. The $\delta^{18}$O values vary between $-7.0$ and 5.0‰ with a mean value of $-6.0 \pm 0.4$ ‰, which
makes them less dispersed than the $\delta^{13}$C record. Nine negative oxygen isotope excursions (NOIEs) are more
        negative than the mean value − 1 SD, amongst which one is below 2 SD reaching a minimum value of $-6.8$‰ at
        19 m. The NOIEs do not correspond with NCIEs described above.

### 4.3 Major and trace elements

        Titanium (Ti), Aluminium (Al) and Zirconium (Zr) concentrations measured on bulk paleosols are plotted in
Figure 5. These elements are commonly considered as immobile and are expected to concentrate in more
        weathered soils. Ti values vary between 0.17 and 0.52% with a mean value of 0.34% and a standard deviation of
        0.08. Al values vary between 3.57 and 9.89% with a mean value of 6.38% and a standard deviation of 1.53. Zr
        values vary between 66 and 203 ppm with a mean value of 127 ppm and a standard deviation of 35. Mean annual
        precipitation (MAP) estimates values vary between 205 and 755 mm/y with a mean value of 387 mm/y and a
standard deviation of 107. Ti, Al, Zr and MAP show a similar trend starting from the base of the section with a
        global increase of all values toward NCIE C and a decrease afterwards. All NCIEs show higher value of Ti, Al,





Zr and MAP except NCIE F. Based on Bull (1991), an average value of 387 mm/y for the MAP in the Chiriveta section represent a semi-arid climate (Fig. 5). All NCIEs show an increase in MAP.

## 5 Discussion

### 5.1 Carbon and oxygen isotopic record

In continental successions, the carbon isotope composition of pedogenic carbonate nodules—which consists of calcareous concretions between 1 mm and 4 cm diameter formed *in situ* in the floodplain—have been proven to reflect global $\delta^{13}C$ variations (Abels et al., 2016; Koch et al., 1992; Schmitz and Pujalte, 2003), and may therefore be considered, sometimes together with the oxygen isotope composition ($\delta^{18}O$), as reliable proxy for

environmental condition occurring during their formation (e.g., Millière et al., 2011a, 2011b). The carbon isotope composition of the soil carbonate nodules depend on the $\delta^{13}C$ value of the soil $CO_2$, which in turn is a function of the $\delta^{13}C$ of the atmospheric $CO_2$, the overlying plants as well as soil respiration (Abels et al., 2012; Bowen et al., 2004; Cerling, 1984).

The $\delta^{13}C$ vs $\delta^{18}O$ diagram for the pedogenic carbonate nodules from the Chiriveta section ($r$ = -0.26, n = 149)

suggests a good preservation of the primary isotopic signal (Figure 6), with an average value of $\delta^{13}C = -7.7 \pm 1.6$ ‰ similar to mid-latitude late-Palaeocene to Eocene continental $\delta^{13}C$ values (excluding the PETM samples) observed elsewhere (e.g., McInerney and Wing, 2011; and references therein), and a spread comparable with $\delta^{13}C$ values from carbonate nodule analysed for the same period in the Bighorn Basin (Bowen et al., 2001). Fig. 6 emphasizes that early-Eocene carbonate nodules display overall more negative $\delta^{13}C$ values than the Holocene

nodules, that is consistent with global data (Zachos et al., 2008). Pre-PETM $\delta^{18}O$ values from carbonate nodules from the same area ($-4.5 \pm 0.4$ ‰) (Hunger, 2018) show similar range than those measured in the Chiriveta section ($-6.0 \pm 0.4$ ‰). Moreover, the $\delta^{18}O$ values of soil carbonates from the Pyrenean foreland basin (excluding the PETM) ($-5.3 \pm 0.9$ ‰) indicate a more coastal-influenced isotopic signature (Cerling, 1984) compared for example to those of the Bighorn Basin ($-9.0 \pm 0.6$ ‰).

A hyperthermal event recorded in marine sediments is defined by a paired negative carbon and oxygen stable isotope excursions that are more negative than the mean value minus 1 SD (Turner et al., 2014). This definition may not be applicable to continental deposits, because continental systems respond differently than marine systems to the carbon cycle perturbations. Indeed, the $\delta^{13}C$ value of the marine dissolved inorganic carbon is influenced by dissolution of carbonates at depth (McInerney and Wing, 2011), whereas $\delta^{13}C$ in

pedogenic nodules vary with soil properties, atmospheric and soil $pCO_2$, the rate and nature of carbon input and/or output by soil respiration (Bowen et al., 2004; Sheldon and Tabor, 2009). These processes may cause a misleading estimation of CIE in soil carbonate nodules and in marine carbonates (McInerney and Wing, 2011). Nevertheless, we used Turner et al. (2014)'s hyperthermal definition as a starting point to filter the high-resolution variations in the Chiriveta section. 16 samples were identified with NCIE values more negative than

the mean $-1$ SD. Among these 16 samples, we recognized 6 discrete NCIEs (named A – F in Fig. 5 and 7). Both marine incursion M1 and M2 show an abrupt shift from $-9$ to $-10$‰ $\delta^{13}C$ values in continental towards more (positive) marine values of $-4$ to $-2$‰; this point to a progressive higher contribution of seawater to the formation of the carbonate nodules.





Six correlation options with the global record were explored in the time-window of the Castissent Fm
(Figure S1 and S2 in the Supplement). Correlation presented in Figure 7A was favoured for the following
reasons: i) it shows reasonable sedimentation rates variations, ii) is coherent with the NCIE amplitude of the
global record, and iii) it yielded the highest correlation coefficient ($r$ = 0.65, n = 71). Moreover, it plots on the
same trend regarding hyperthermal NCIE amplitudes in marine and continental environments suggesting a
similar isotopic dynamic as events I1, I2, H2 and ETM2 (Figure 7B). Based on these observations and the
resulting correlation, we suggest that only hyperthermal U is preserved in the Chiriveta section and that it is
correlated with NCIE D. Sedimentation rate obtained with the favoured correlation (Figure 7) varies between
0.1-0.29 mm/y and the correlation coefficient of $r$ = 0.65 suggests an overall good signal preservation in the
studied continental section for a 40 ky climatic event.

Soil organic matter sensitivity to a change in temperature is critical concerning today's global warming
(Melillo et al., 2014; Trumbore et al., 2006), because it represents two-thirds of the terrestrial carbon pool and
contains twice as much carbon as atmospheric $CO_2$ pool (Carrillo et al., 2018; Schimel et al., 1994). An increase
in temperature could therefore potentially release important amount of $CO_2$ into the atmosphere (Trumbore et al.,
2006). As the amplitude and duration of Eocene NCIEs are approximately 30% of the ones recorded in the
PETM, we hypothesize that the climatic effects of smaller-scale hyperthermals can be linearly scaled to the
PETM. Based on this assumption and in order to get a rough approximation without considering a non-linear
sensitivity response, a smaller-scale hyperthermal would imply a release of approximately 500 to 1500 Gt of
carbon to the ocean and atmosphere reservoir and a global temperature rise of about 1.5−2.5° C. This estimation
correspond to the 1500 − 4500 Gt of carbon released during the PETM, causing a rise of 5−8°C (Bowen et al.,
2006), and is in line with previous estimations of ~3 and ~2°C warming for ETM2/H1 and H2 events
respectively (Stap et al., 2010).

The $\delta^{13}$C mean value in the Chiriveta section is −7.7 ± 1.6 ‰. This value reflects an overall equilibrium
with a mean atmospheric $CO_2$ of −7‰ (Koch et al., 1995) and is coherent with pre-PETM $\delta^{13}$C values of −7.1 ±
0.9 ‰ found in the same area (Hunger, 2018; Fig. 6). It is possible to calculate from the (small-scale)
hyperthermal $\delta^{13}$C excursions in the marine environment the shift to be expected in soil carbonate nodules by
using known fractionation coefficients (Koch et al., 1995, 2003); the expected $\delta^{13}$C value in carbonate nodules,
only considering the respiration of organic matter, is of −11‰ (Fig. 8). This value is within the range of those
measured in Chiriveta section, where some nodules reach values as low as -10.9‰. We suggest that the bacterial
respiration of organic matter, enhanced by warmer temperatures (e.g.; Davidson and Janssens, 2006; Trumbore
et al., 2006), may also have contributed to the lower $\delta^{13}$C values of nodules during the NCIEs (Fig. 8). On
geological timescales, soil organic carbon can be considered at steady state with equal organic carbon inputs and
outputs from the soil (Koven et al., 2017). Respiration (carbon output after mineralization as $CO_2$) is thought to
be more sensitive to global warming than gross primary productivity (organic carbon input as organic matter)
leading to a depletion of the total soil carbon pool with time during transient global warming events; although
the precise sensitivity of gross primary productivity remains poorly constrained (Davidson and Janssens, 2006).
Large uncertainties remain about carbon dynamics and their timescale in the soils during climate changes.
Parameters such as the vegetation type (Klemmedson, 1989), temperatures (Koven et al., 2017), soil
geochemistry (Doetterl et al., 2015; Torn et al., 1997), and soil water content (Davidson et al., 2000) have been
shown to be important controlling factors within historical timescales. Considering these caveats, we estimate





the maximum possible contribution of enhanced soil carbon respiration to negative $\delta^{13}C$ excursions during the
NCIEs. Using typical values for the organic carbon reservoir comprising fast and slow cycling carbon in soils in
arid to semi-arid ecosystems of 5.6–19.2 kgC/m² (Klemmedson, 1989; Raich and Schlesinger, 1992), respiration
fluxes starting at a steady state value of 0.5 kgC/y, and a respiration rate sensitivity *ca.* 5%/degree (Raich and
Schlesinger, 1992), we estimate that all of the organic carbon in soils would be consumed within 250 to 850 y,
given an increase of 1°C and without changing the carbon input rate. Though there are a number of assumptions
in this first-order estimate, the timescale of soil carbon depletion is substantially shorter than our estimate of the
timescale of the NCIE (~36 ky) (Fig. 7).  As evidenced by this calculation, an increase in soil respiration
triggered by warmer temperatures cannot be the sole mechanism driving the NCIE shift over multi-millennial
time-scales. Instead, we suggest that during these transient warmings, this mechanism is associated with a high
primary productivity—resulting in a greater input of carbon to the soil—leading to an overall higher soil
respiration of organic matter. Coupled with lower atmospheric $\delta^{13}C$ during hyperthermals, this mechanism
caused a pronounced NCIE in soil carbonate nodules.

### 5.2 Geochemical signature of hyperthermal events

Major and trace elements compositions of floodplain sediments is a function of river dynamics, climate, and
sediment grain-size (Lupker et al., 2012; Turner et al., 2015). Based on the NCIE described above, we defined
six intervals in the Chiriveta section. Each interval shows a relative enrichment (up 10 to 30% compared to the
average value) in immobile elements such as Ti, Al and Zr (Fig. 5). To ensure that major and trace
concentrations are not grain-size biased, we plotted grain-size proxies Si/Al, Ti/Al and Zr/Al (Lupker et al.,
2012; Turner et al., 2015), which all exhibit a relatively stable trend, not correlated with the immobile element
concentrations (Figure S3 in the Supplement). The enrichments in Ti, Al and Zr suggest mature paleosols with
potential intense weathering due to enhanced humid climatic conditions; but may also correspond to a longer
exposure time on a stable floodplain, allowing leaching of mobile elements and relative enrichment of immobile
elements (Sheldon and Tabor, 2009). Pedogenic nodules are frequent in well-drained soil profiles associated
with a climate regime where the potential evapotranspiration is greater than the mean annual precipitation rate
(Slessarev et al., 2016) and with a mean annual precipitation < 800 mm/year (Cerling, 1984; Retallack, 1994;
Sheldon and Tabor, 2009). These conditions correspond to climate ranging from arid to sub-humid conditions
(Hasiotis, 2004; Hyland and Sheldon, 2013; Prochnow et al., 2006). This agrees with MAP values obtained for
the paleo-precipitation estimate (Fig. 5) and with a smectite/kaolinite >1 assemblage dominating some of the
studied soils (Nicolaides, 2017, Table S1 in the Supplement); all fitting well with a semi-arid to sub-humid
contrasted climate with seasonal humidity (Arostegi et al., 2011). Associated with NCIEs C and D in red bed
deposits, sub-milimetric iron-oxide and hydroxides nodules made of concentric hematite and goethite were
found together with carbonate nodules. This suggest a contrasted climate as hematite forms under more arid soil
condition than goethite (Kraus and Riggins, 2007). Together, these observations are in line with an acceleration
of the hydrological cycle and a higher seasonality already observed during the PETM, H1, H2; I1 and I2
hyperthermals (Bowen et al., 2004; Dunkley Jones et al., 2018; Nicolo et al., 2007; Slotnick et al., 2012).
Therefore, combined with NCIEs, we suggest that small scale hyperthermals in continental records can be
recognized by an increase in the weathering index (Hessler et al., 2017) and by an increase in the immobile
element concentrations,  both related to an increase in precipitation intensity.



### 5.3 High-resolution hyperthermal signal

The high-resolution isotopic and elemental record of the Chiriveta profile allow us to highlight the dynamics and
variability of a hyperthermal event. We do not observe a unique peak in $\delta^{13}C$, but rather a stepped isotopic signal
suggesting, together with above-discussed geochemical data, a climatic oscillation alternating with variably
intense precipitations and leaching conditions during a climax spanning *ca.* 150 kyr (interval NCIE B to D).
Such a climatic behaviour, already described during the pre-onset PETM excursion (Bowen et al., 2014), may
indicate a back and forth climatic response to carbon cycle perturbations. Moreover, the $\delta^{13}C$ climax (NCIE D)
does correspond neither to the highest concentrations of immobile elements nor maximum MAP estimates
happening during NCIE C, since it predates from *ca.* 50 kyr the NCIE D (Fig. 7). The minimum $\delta^{13}C$ value
therefore does not seem to be coeval with the most extreme climatic response, suggesting a complex
environmental response. However, because sedimentation in floodplain depositional settings is a function of the
channel position and flood frequency, the relative concentration of elements only likely reflects the changes in
river dynamics instead of climatic variability, which could explain the mismatch between minimum values in
NCIE and the climatic response. More high-resolution hyperthermal studies in coeval continental sections are
needed to better understand the relationships between proxies.

### 5.4 Preservation potential of hyperthermals in continental sections

Since the first studies that applied sequence stratigraphy concepts onto continental deposits, the preservation of
environmental signals in the continental stratigraphic record has been considered incomplete, especially during
falling sea-level (Shanley and McCabe, 1994; Wright and Marriott, 1993). Even at higher-resolution timescales,
floodplain deposits are still considered as fragmentary and discontinuous in nature due to non-continuous flood,
avulsion, and channel migration sedimentation processes and the irregular depositional thickness relative to the
position of the channel (Turner et al., 2015). This potential incompleteness of the sedimentary record (Barrell,
1917; Sadler, 1981) and the capacity of a sedimentary section to document a continuous paleoclimatic signal has
probably led many workers to prefer the deep-marine records. Major events such as the PETM event has proven
to be detectable in both marine and continental environments (e.g.; Abels et al., 2016; Koch et al., 1992), but the
signal and preservation potential of smaller scale climatic events (e.g. hyperthermal events L to W in Lauretano
et al., 2016), is somewhat uncertain (Foreman and Straub, 2017).
Consequently, to assess in a quantitative matter the preservation potential of a hyperthermal event with a generic
40 kyr duration (Sexton et al., 2011), we calculated the compensation time scale (Tc). Tc is a characteristic time-
scale in an alluvial basin below which stratigraphic signals with shorter durations may be of autogenic origin,
thereby giving a scale below which allogenic forcing should be interpreted carefully (Foreman and Straub, 2017;
Trampush et al., 2017; Wang et al., 2011). In other words, an external forcing signal with a duration smaller than
Tc will be challenging to identify from background variability; the external forcing must be therefore of a longer
duration than Tc and optimally twice Tc (Foreman and Straub, 2017). Tc max can be calculated by dividing the
topographic roughness or maximum channel depth by the average subsidence or deposition rate (Wang et al.,
2011). Based on preserved channel fills in La Roca and Chiriveta sections, we estimated a maximum channel
depth to be 6 m with an average of 3.75 m (fitting previous measurements of maximum 7 m by Nijman and
Puigdefabregas (1978)). With a thickness of 150 m for the Castissent Fm in the La Roca section (Marzo et al.,
1988) and of 101 m in the Chiriveta section (this study) and using the maximum and minimum age extension of



the Formation, we obtain sedimentation rates between 0.1 and 0.29 mm/yr. These values are within sedimentation rates of Eocene floodplain succession (Kraus and Aslan, 1993). Using an average sedimentation rate of 0.17 m/kyr, we obtained a mean Tc of 22,000 yrs. A hyperthermal event of 40 kyr, being approximately

twice as long as the estimated Tc, should be recorded.

There are five Eocene hyperthermal events (PETM, ETM2/ELMO/H1, H2, I1, I2) identified worldwide (e.g.; Abels et al., 2016; Schmitz and Pujalte, 2003). If we add the data from this study (U event), we can estimate a mean thickness for hyperthermal events of *ca.* 27 m for continental and 1.3 m for deep-marine succession, respectively (Abels et al., 2016; Bowen et al., 2001; Lauretano et al., 2015; Lourens et al., 2005; Nicolo et al.,

2007; Schmitz and Pujalte, 2003; Slotnick et al., 2012; Westerhold et al., 2018). Thus, regarding the hitherto studied sections, terrestrial strata recording hyperthermals are potentially one order of magnitude thicker with resolutions likely higher than deep-marine sections. Therefore, continental strata, which are directly linked to environmental conditions occurring at the time of their formation, might preserve a better record of past climatic events (Sheldon and Tabor, 2009). The continental record of past climatic events might have been overlooked; if

such record can be proved complete, the potential climatic events preservation is higher and likely of high resolution.

## 6 Conclusions

A new high-resolution isotopic record from the paleosol-rich Chiriveta section succession allows to identify a prominent negative carbon isotope excursion (NCIE) in continental deposits that we suggest to be the "U" event,

providing new insights into the climate and carbon cycle dynamics during a hyperthermal event. This climatic event, identified for the first time in continental deposits, reaches $\delta^{13}C$ values of 2 sigma (standard deviation) below the mean and is preceded and followed by several smaller-scale stepped NCIEs, which are interpreted as moments of enhanced primary productivity, leading to an overall higher soil respiration. We show that all these NCIEs are relatively enriched in immobile elements (i.e., Ti, Zr and Al) and display an increase in MAP

estimates. These observations coupled with the presence of iron-oxide nodules on an overall weathered succession, suggest a contrasted climate and an increase in precipitation rates during these events. The data presented in this study suggests a period of *ca.* 150 kyr of oscillating climate alternating average and above background weathering conditions. Finally, the results of this study provide support to the recognition and importance of hyperthermal events in continental successions as well as in the preservation potential of such

deposits.

**Acknowledgements.** The authors would like to acknowledge the lifetime work of Josep Serra Kiel, whose research and scientific contributions in the Pyrenees have been fundamental to this work and much beyond. This study has benefited from scientific discussion and field work with M. Perret, A. Nowak, C. Läuchli, T. Hunger,

J. Vernier and T. Maeder. Work supported by an Augustin Lombard grant from the SPHN Society of Geneva. Field work was supported by an Equinor grant to SC.

Isotopic, majors and trace data can be found in the supporting information (Table S2 in the Supplement)



**Author contributions.** LH led the field work, sampling, sample preparation, data interpretation and writing. TA contributed to field work, sampling, data interpretation, discussion and writing. JES performed stable isotope analysis, data interpretation and writing. JKR interpreted the data and writing. MPM and EC contributed to fieldwork, sampling, discussion and writing. CP, JC and AF supervised the fieldwork, discussions and writing. EV led discussions on the paleosols. KK and MH performed the XRF analysis. SC supervised the project and writing.

The authors declare that they have no conflict of interest

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



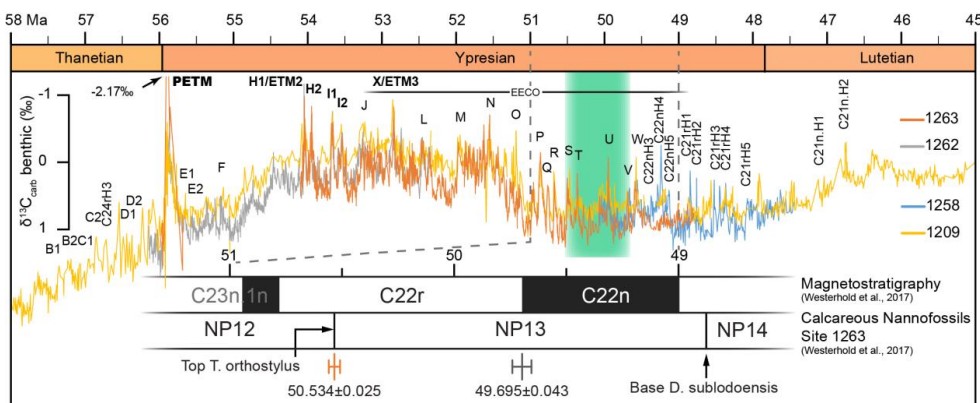

**Figure 1: Late Paleocene and early-Eocene benthic carbon isotope record from Sites 1209, 1258, 1262 and 1263. Top of Chron C22r and top of T. orthostylus zone from site 1263 from Westerhold et al. (2017). Hyperthermal nomenclature from Lauretano et al. (2016) and Westerhold et al. (2017). Castissent Fm extension in green.**

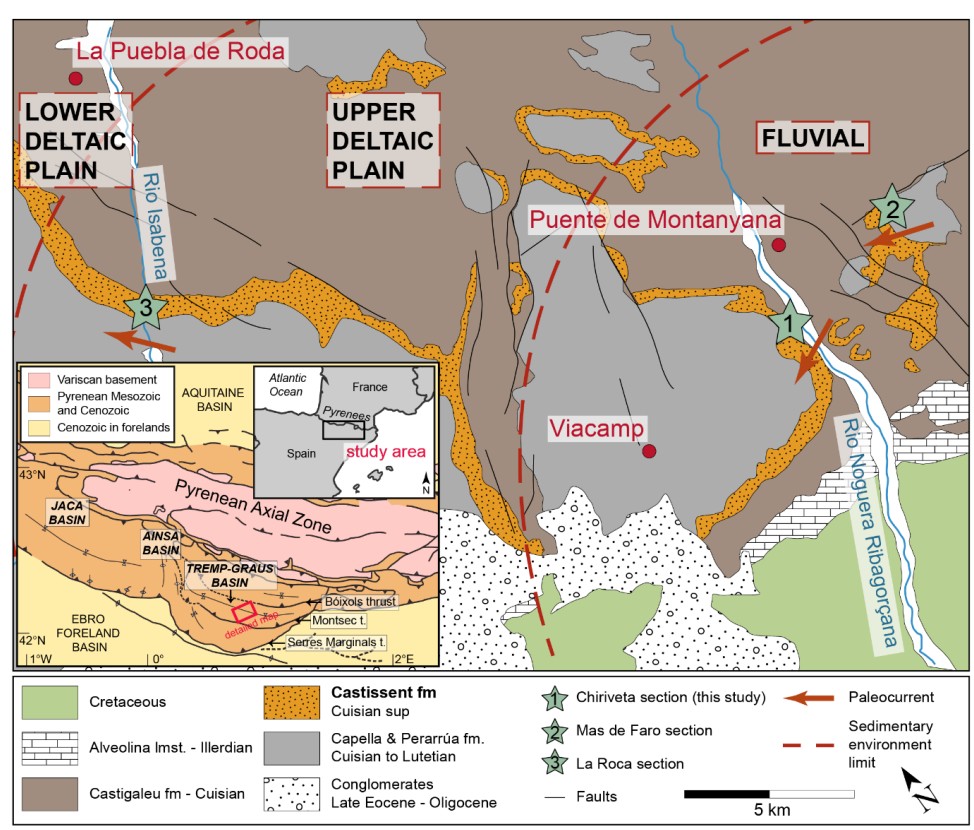

**Figure 2: Simplified situation and geological map of the study area with main depositional paleo-environments (e.g., Nijman, 1998). The Castissent Fm is a prominent fluvial unit particularly well-exposed in the Noguera Ribagorçana and Isabena river valleys. (1) Chiriveta section (2) Mas de Faro (3) La Roca section. Main paleoflow directions indicated in orange (from Nijman and Puigdefabregas, 1978). Regional map after Teixell (1998).**





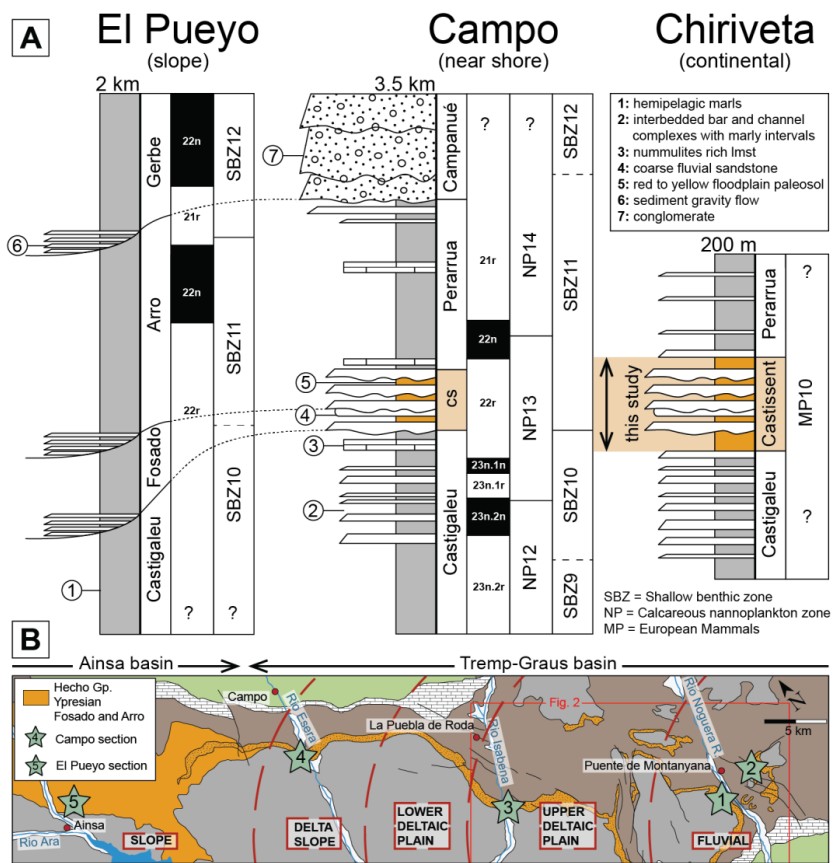

**Figure 3: A - Time constraints on the Castissent Fm. MP zone from the continental section from Checa-Soler (2004) and Payros and Tosquella (2009). SBZ and NP in the Campo section from (Kapellos and Schaub, 1973; Schaub, 1966, 1981; Tosquella, 1995), magnetostratigraphy from Bentham and Burbank (1996). SBZ in El Pueyo section from Payros and Tosquella (2009). Magnetostratigraphy in El Pueyo from Poyatos-Moré (2014). B – Extended map of the study area. For map legend and references, see Fig. 2.**

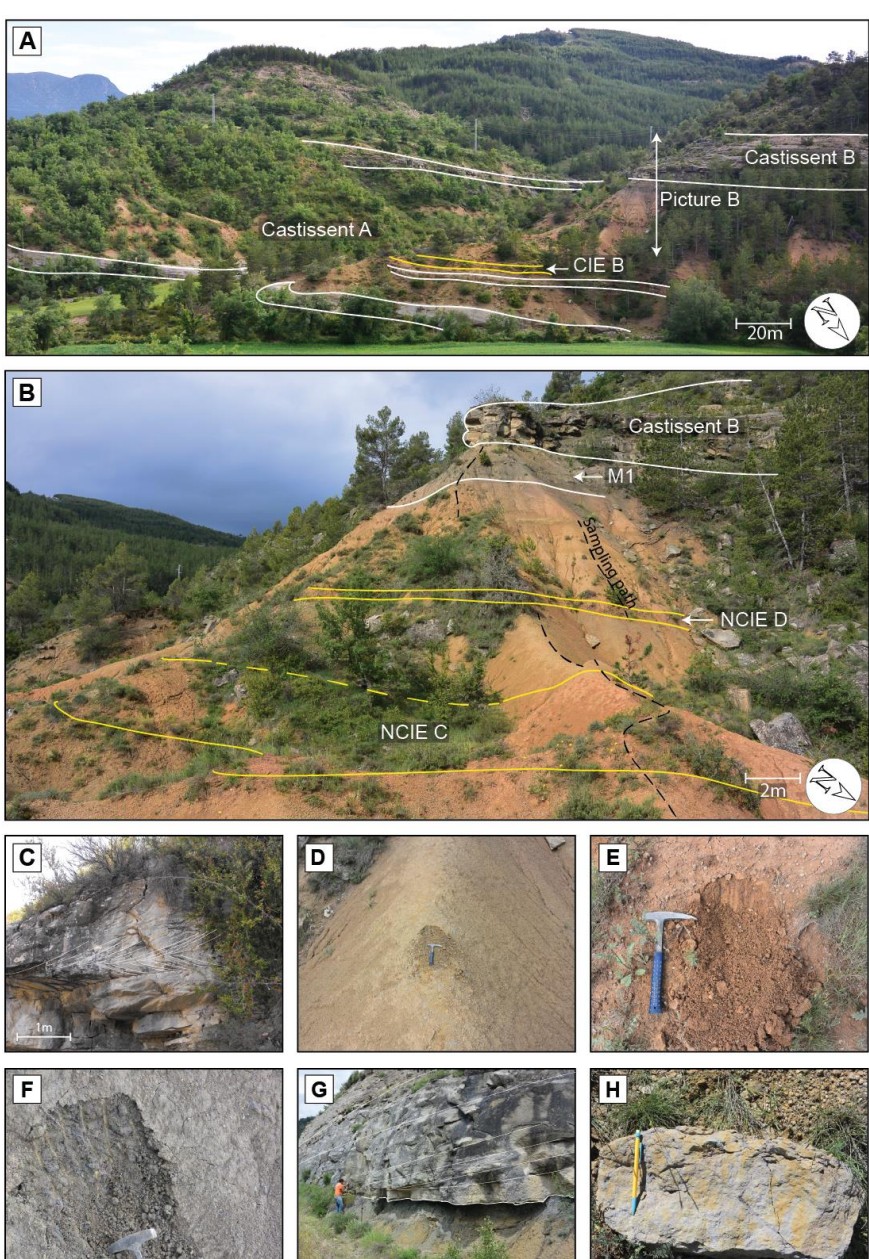

**Figure 4: Field images of the Chiriveta section (42°7'56.57''N, 0°41'19.45''E). A – Outcrop view of Members A and B of the Castissent Formation. B – Close-up view of the upper part of Castissent A Member. Fluvial channel-fill deposits, intercalated in reddish floodplain and overbank deposits and regional marine incursions (M1). C – M0, first marine incursion at the base of the Castissent Fm described by Marzo et al. (1988) expressed in the Chiriveta section by a tidal-influenced coarse sandstone with herringbone cross-stratification. D – Yellow mottled paleosol between NCIE C and D. E – Redfloodplain interval equivalent of the NCIE C. F – 2 m-thick grey interval interpreted as poorly drained brackish water facies and equivalent to the marine incursion M1. G – ~6m thick laterally extensive Castissent B sandbody incised in the underlying floodplain deposits. H – Mottled silt, interpreted as pedogenetic fluvial channel overbank deposits.**



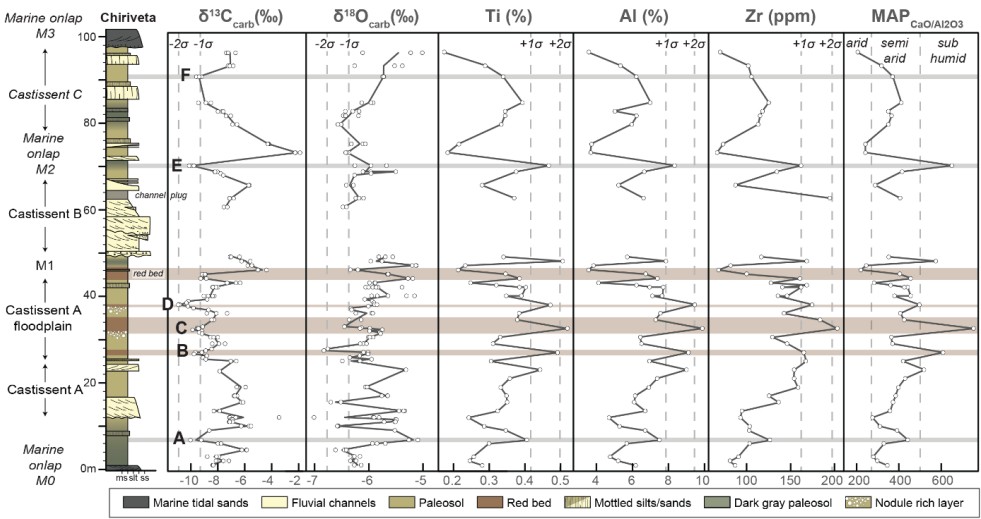

**Figure 5: Isotopic and geochemical data from the Chiriveta section. For the isotope dataset, the curves passes through the mean values at each sample position. Samples with minimum in $\delta^{13}$C values below 1 and 2 standard deviation are labelled A to F. Mean Annual Precipitation (MAP) was estimated from the empirical relationship between MAP and CaO to Al$_2$O$_3$ ratio (Sheldon et al., 2002).**

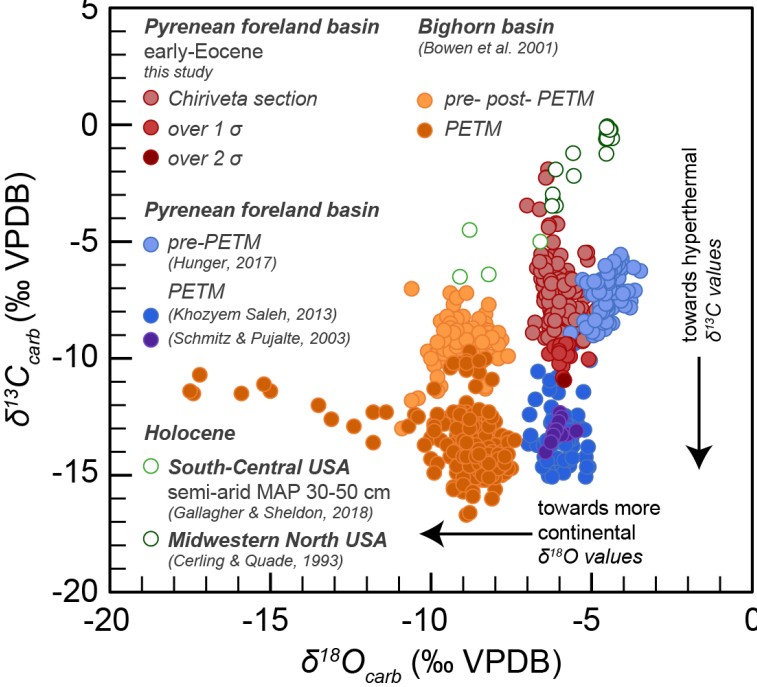

**Figure 6: Continental $\delta^{13}$C and $\delta^{18}$O values from the early Eocene Castissent Fm in the Chiriveta section (this study) plotted with pre- and syn-PETM $\delta^{13}$C and $\delta^{18}$O values from the same area (Hunger, 2018; Khozyem Saleh, 2013) and Pre-, syn and post-PETM values from the Bighorn Basin (Bowen et al., 2001) as well as recent pedogenic carbonate isotopic values (Cerling and Quade, 1993; Gallagher and Sheldon, 2016).**





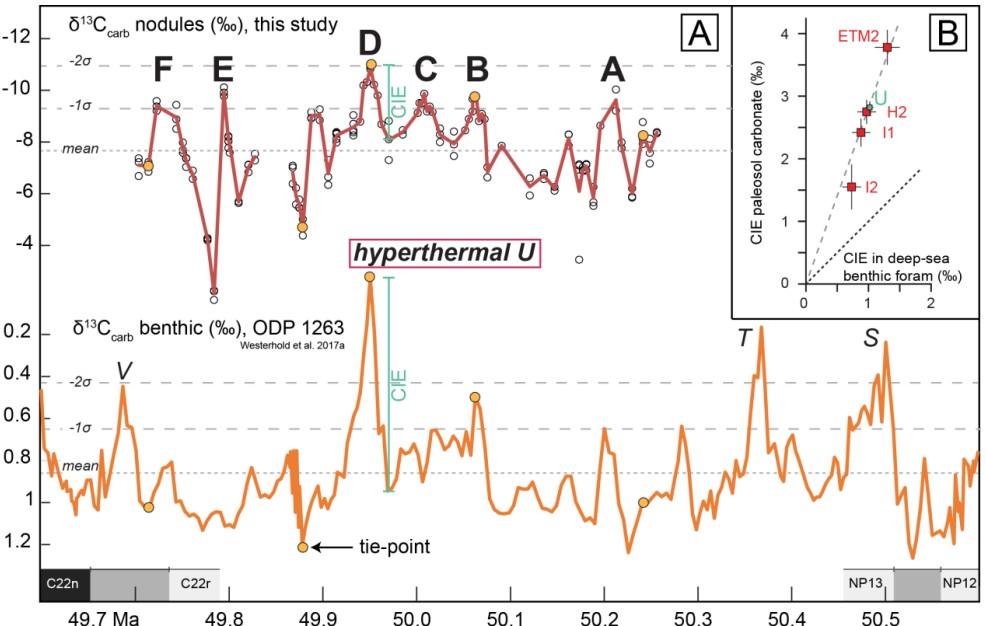

**Figure 7:** A - Scaling of the Chiriveta isotopic section with the time equivalent interval of site 1263 (Westerhold et al., 2017). The correlation was calculated using the Analyseries software (Paillard et al., 1996) and centred on NCIE D and hyperthermal U. Mean, minus 1 and 2 SD lines on the global record were calculated sets over the selected time period. The correlation coefficient (*r*) between the two curves is 0.65. B - Hyperthermal U amplitude in paleosol carbonate and benthic foraminifera (inset B after Abels et al. (2016))

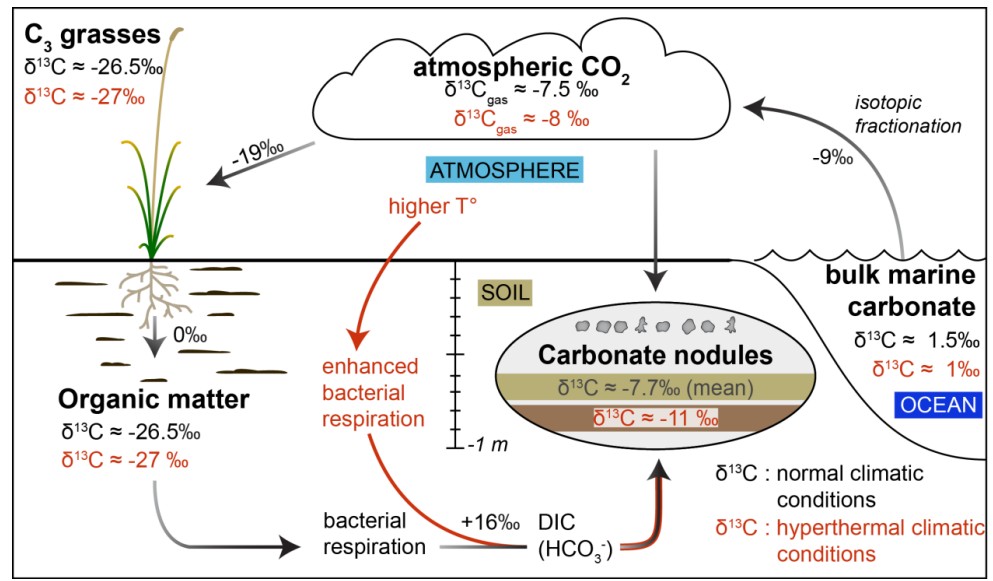

**Figure 8:** Components influencing the *δ*¹³C values of pedogenic carbonate nodules. Mean early Eocene bulk marine carbonate and small scale hyperthermal (all except PETM) are from Westerhold et al. (2018). Fractionation value between organic matter and carbonate nodules are based on Sheldon and Tabor (2009). All other fractionation values are based on Koch et al. (1995). Mean carbonate nodule values come from this study.