# Peer review of "Alluvial record of an early Eocene hyperthermal within the Castissent Formation, Pyrenees, Spain"

_Climate of the Past, 2019_

## Referee Comment (RC1) · Thomas Westerhold (Referee) · 18 Sep 2019

The authors present new stable carbon isotope data measured on pedogenic carbonate nodules from a terrestrial fluvial sedimentary succession from the early Eocene Castissent Formation in the South-Central Pyrenees. Their endeavor is the important task to correlate marine and terrestrial realm during the Early Eocene Climate Optimum, the warmest time on the planet in the Cenozoic characterized by frequent transient global warming events. This study is highly relevant because regional terrestrial archives are needed to understand reginal responses to global warming events, in particular monitoring changes in the hydrological cycle. In essence the new record provides evidence that hyperthermal events in magnetochron C22r are associated with regional increase in precipitation intensity. It is well suited to be published in CoP after

minor revisions.

The manuscript is well written, clearly structured, has superb figures and informing supplementary section. Here I particularly comment the age model, the soil nodule carbon dynamics and their implications has to be judged by someone else. Correlation between marine (lower sedimentation rates) and terrestrial (high sedimentation rates) is notoriously difficult. The author present a robust and clear approach to get a decent age model comparable to the marine records. Uncertainties and different correlation options are discussed giving the opportunity to evaluate the effects of various age models. Supported by wider correlation to nearby outcrops and bio-magneto-stratigraphic data to me the preferred correlation makes perfect sense. Also because the sedimentation rates of this option is in line what has been observed in similar settings like the Bighorn Basin. The only concern is the scaling of the U labeled hyperthermal in the deep sea and the CIE D in the carbonate nodules of the Chiriveta record. How do the events labeled B and C plot in Figure S2? However, the overall pattern in the carbon isotope records of marine and the Chiriveta record match well reinforcing the age model preferred.

The manuscript hopefully will stimulate seeking for more continental records, although the age model construction will be difficult. Despite this, the section 5.4 Preservation potential of hyperthermals in continental sections is informing but a bit out of context. It is clear to the community that higher sedimentation rates allow a more detail insight. This section, if kept in the manuscript, also needs to discuss that sedimentation is not uniform (steady) in terrestrial records but highly dynamic (50m away from the section things will look very different, see Bighorn Basin Project results where outcrops studies and drill cores allow a 3D view).

Comments: Abstract Line 23 – Hyperthermal cannot be "potential analogues, in the geological record, to the ongoing anthropogenic modification of global climate". Background conditions 50+ million years ago were much different. But the events can help to test the assumptions made by climate models and revise them for a better understanding of the climate system dynamics.

Line 44-45: remove "towards icehouse conditions eventually reached later in the Cenozoic"

Line 48: "Turner et al., 2014" change to Kirtland-Turner et al. 2014 in the entire text

Line 51: "e.g., Westerhold et al., 2018"; add Lourens et al. 2005, Sexton et al. 2011, Kirtland-Turner et al. 2014, Lauretano et al. 2015, 2016. They published the records used.

Line 52: "Early Thermal Maximum (ETM) 2, H2, I1, I2, and ETM3/X events"– correct to Eocene Thermal Maximum. Change the wording of the sentence to clarify the nomenclature (ETM is only for 1,2,3; H1 H2 etc. are from Cramer et al. 2003 revised by Lauretano..., Sexton et al. 2011 suggested the relative to magnetochron scheme, see Westerhold et al.). In Figure 1 the text refers to this, please streamline the manuscript text accordingly.

Line 54: "In the stratigraphic record, these events are primarily characterized by important negative carbon isotope excursions (NCIEs)" – rephrase!, they are characterized by a paired negative excursion in carbon and oxygen isotope data; do not use NCIE throughout as it confuses with the commonly used CIE (Carbon Isotope Excursion) abbreviation for the e.g. PETM.

Line 150: please specify the target material of the X-ray tubes (Mo, Rh?).

Line 180: provide tie-points to ODP 1263 as a table in the supplement and add the age to your data tables of isotope as well as XRF results.

One important thing would be to show images of the soil nodules from the Chiriveta record in the supplement.

---

## Referee Comment (RC2) · Anonymous Referee #2 · 20 Sep 2019

Interactive comment on the manuscript "Alluvial record of an early Eocene hyperthermal, Castissent Formation, Pyrenees, Spain" by Louis Honegger et al,

Louis Honegger et al claim to have located for the first time in a terrestrial succession the late Ypresian hyperthermal event coded "U", specifically within the Castissent Formation, a well-known alluvial-fluviatile unit of the southern Pyrenees. Their claim is based on a stable carbon isotope profile obtained from pedogenic carbonate nodules of a single section (Chiriveta). Their study, if correct, is potentially important, because to fully understand the effects of hyperthermals events their impacts on different settings need to be investigated. Howevr, although they might be correct in their claim, I am not entirely convinced that the fact presented in the manuscript incontrovertibly demonstrate that this is the case. Besides, some items of the manuscript are, in my

view, somewhat confuse. Below I list the main points of my concern:

1. The title of the manuscript (Alluvial record of an early Eocene hyperthermal, Castissent Formation, Pyrenees, Spain) is misleading: When I first read it I though the authors meant that the Castissent Fm had resulted from the effects of an hyperthermal – which would be impossible. I therefore suggest something like: Alluvial record of an early Eocene hyperthermal within the Castissent Formation, Pyrenees, Spain.

2. In lines 36−40 they state that "The results show that even relatively small-scale hyperthermals compared with their prominent counterparts, such as PETM, ETM2 and 3, have left a recognizable trace in the stratigraphic record...". The environmental effects of the above-mentioned hyperthermals in terrestrial setting, especially those of the PETM, are very prominent (e.g., Foreman et al, 2012), a fact that made their record easily recognizable in the field. For the alleged U event (NCIE D) the information provided does not justify the above assertion. Thus, in Fig. 7 the event seems to be represented by a comparatively thin interval (<1 m), whereas the previous NCIE C, which is of smaller magnitude, is recorded by a ∼3 m thick interval. A better documentation of the sedimentological features of the alleged U event is needed, as it is the focus of the study. It would also help if such (distinctive ?) features could be observed in other section(s).

3. Constraint on the age of the Castissent Fm is somewhat vague. It is not based on data from the Chiriveta section itself, but on bio- and magnetostratigraphic studies of previous authors (Kapellos and Schaub , 1973; Bentham and Burbank, 1996), carried out in the Campo section, 40km westward. Based on them the authors indicate that the Castissent Fm occurs within the D. lodoensis nannoplankton zone (= NP 13), with the base and top of the nannozone being respectively situated at ca 200 m below the base of the Castissent, and at ca 100 m above its top. My doubts about the reliability of the Kapellos and Schaub zonation (1973) partly stem from the fact that shallow marine facies such as those of the Campo section are not favorable for the preservation of nannofossils, and that therefore are not entirely reliable. The NP9/NP10 boundary

provides a proof of this, for K & Sch′73 did situate it ABOVE the so-called Alvelina limestone, while Orue-Etxebarria et al. (2001; Marine Micropaleontology 41, 45−71) proved that it occurred BELOW such unit, a finding that permitted to correctly place the PETM interval in the Campo section (see Fig. 1 of this Comment). More to the point, as shown also in Fig 1, the location of the top NP 13 zone is somewhat ambiguous: K & Sch′73 state in their text that the NP 13 zone spans from km 58.6 (base) to km 56 (top), whereas in their columnar section the top of the zone is placed at sample 32. Such uncertainty raises doubts about the magnetostratigraphic calibration of Bentham and Burbank (1996), likely based on the K & Sch′73. Indeed, in Fig. 3 of the manuscript the NP13/14 boundary is placed within the C22n magnetozone, whereas in Fig. 1 (from Westerhold et al, 2017) is located within C23r.

4. The completeness of the studied section is debatable In the first paragraphs of chapter 5.4 ("Preservation potential of hyperthermals in continental sections"), the authors acknowledge that alluvial-fluvial stratigraphic records are considered incomplete by many authors (e.g., Shanley and McCabe, 1994; Wright and Marriott, 1993; Turner et al., 2015; Barrell, 1917; Sadler, 1981). In the present case, Marzo et al. (1988) concluded that "The sedimentation of the Castissent Formation was structurally controlled by an interplay of vertical basement movement due to thrust stacking in the hinterland and surficial thrust displacement to the foreland resulting in alternating southward and northward shift of the fluvial system". The Chiriveta section is close to the foreland thrust (Montsec thrust) and, in such dynamic scenario, it is doubtful that it would have accumulated a (near) continuous succession. But, even if that were the case, it seems rather improbable that the section would be complete enough to have recorded ALL the minor NCIES detected in the ODP 1263 site, as shown in Fig. 7.

5. Section 4.1 of the manuscript ("Overview 0f the Castissent Fm at the Chiriveta section) seems to be misplaced. I suggest to remove it from the Results section and place it after the Chapter 2, Geological setting.

6. I have not had the time to check out all the references, but in a quick glance I

can point out that some of them are incomplete: Hunger, T.: Climatic signals in the Paleocene fluvial formation of the Tremp-Graus Basin, Pyrenees, Spain. University of Geneva., 2018. Is that a Thesis? How many pages? It is published or unpublished?

Poyatos-Moré, M.: Physical Stratigraphy and Facies Analysis of the Castissent Tecto-Sedimentary Unit., 2014. Is that a Thesis? If so, from which University? How many pages? It is published or unpublished?

The list of authors of the reference "Payros, A. and Tosquella, J.: Filling the North European Early/Middle Eocene (Ypresian/Lutetian) boundary gap: insights from the Pyrenean continental to deep-marine record, Palaeogeogr. Palaeoclimatol. Palaeoecol., 280, 313–332, doi:10.1016/j.palaeo.2009.06.018, 2009" is incomplete. Either include all the authors (Payros, A., Tosquella, J., Bernaola, G., Dinarès-Turell, J., Orue-Etxebarria, X., and Pujalte, V.,), or quote it as Payros, A., Tosquella, J, et al.

7. Some previous papers should be referenced. In lines 60−63 the manuscript states that "In coastal marine sections, Early Eocene hyperthermal events are generally associated with an enhanced flux of terrigenous material, interpreted as linked to accelerated hydrological cycle and higher seasonality (Bowen et al., 2004; Dunkley Jones et al., 2018; Nicolo et al., 2007; Payros et al., 2015; Slotnick et al., 2012)..." To my knowledge, one of the first paper pointing out this fact was: Schmitz, B., Pujalte, V., Núñez-Betelu, K., 2001. Climate and sea-level perturbations during the Initial Eocene Thermal Maximum: evidence from siliciclastic units in the Basque Basin (Ermua, Zumaia and Trabakua Pass), northern Spain. Palaeogeogr. Palaeoclimatol. Palaeoecol. 165, 299–320

In lines 63−65 the manuscript states that "Several studies document a spatially heterogeneous hydrological climatic response during the PETM (Bolle and Adatte, 2001; Carmichael et al. 2017; Kraus and Riggins, 2007)". The paper by Giusberti, L., Boscolo Galazzo, F., Thomas, E., 2016. Variability in climate and productivity during the Paleocene–Eocene Thermal Maximum in the western Tethys (Forada section). Clim.

Past 12, 213–240, should be acknowledged, as their compilation made evident such climatic variability.

[Figure]

[Figure]

Fɪɢ. 7. — Esquisse topographique de la coupe paléog de Campo.

Fɪɢ. 8. — Profil schématisé et biozones du Paléogène au S de Campo.

Figure 1, tentative location of the NP 13 on the road map of the Campo section (left), and in the columnar section (right), superimposed on figures 7 and 8 of Kapellos and Schaub. Alternative locations of the NP9/NP10 are also indicated in the columnar section

**Fig. 1.**

---

## Referee Comment (RC3) · Anonymous Referee #3 · 1 Oct 2019

This paper presents a detailed carbon/oxygen isotopic and geochemical dataset from the Castissent Formation to investigate the terrestrial record of early Eocene hyperthermal events. Based carbon isotope stratigraphy, the authors identify 6 hyperthermal events in the Castissent Formation. Using orbitally tuned stratigraphic data tied to mammal biostratigraphic ages, the authors correlate the most prominent of these to Hyperthermal U, previously identified by benthic oxygen isotope stratigraphy from ODP 1263. The authors also use major and trace elements to suggest that hyperthermal events are recorded in terrestrial deposits as horizons with dramatically increased mean annual precipitation proxies. Perhaps the most important conclusion of this paper is that such events can be recognized in the terrestrial record and may improve our understanding of the terrestrial effects of these climate fluctuations.

[Figure]

Overall, this paper presents a large dataset and strong conclusions, and it will be of interest to a broad audience. A few minor questions/recommendations that might improve the paper are presented below:

1) I think this paper would benefit from a more detailed description of the model used to constrain the age of these deposits. Was the placement of the Castissent Formation within European Mammal Zone MP10 based on the same outcrops sampled here, and if not, what is the proximity of that site? Although the authors state that many of the well-dated sections within the Castissent Formation can be physically correlated to the current study area, how can the authors be confident that these are not time-transgressive deposits? Finally, I am skeptical that the age designation bracket of 50.534±0.025 and 49.695±0.043 Ma can be realistically applied to this unit. That extremely precise age range is based on orbital tuning of a marine record, correlated to a continental record, correlated to the current study area. I am not disputing the correlation, just that the precision of the marine record might not be retained through two iterations of lithostratigraphic correlation.

2) The authors note that unlike most marine hyperthermal records, the oxygen and carbon isotopic records are not coupled in the Castissent Formation (the oxygen does not reflect hyperthermal events, whereas the carbon does). Why might this be? Is there evidence of isotopic resetting of the O system (petrographic or other)? How deeply have these rocks been buried? This seems to suggest that even in well-preserved systems, oxygen isotopic records should be used and viewed with caution.

---

## Author Comment (AC1) · 7 Nov 2019

**Response to "Review CP-2019-88 Honegger et al."**

Louis Honegger et al.

*In this response, the original comments are in black and responses by the authors to the reviews are written in blue. Changes in the manuscript are written in red.*

*We thank the reviewer for his work on and appreciation of our study. We greatly value his expertise and views on our study because his work provided some of the motivation for the research questions we are currently pursuing. We answer below point by point to each item raised.*

The only concern is the scaling of the U labeled hyperthermal in the deep sea and the CIE D in the carbonate nodules of the Chiriveta record. How do the events labeled B and C plot in Figure S2? However, the overall pattern in the carbon isotope records of marine and the Chiriveta record match well reinforcing the age model preferred.

*Below, please find the modified figure incorporating in inset B, CIE B, C and D plotted regarding the CIE amplitude in soil nodules and in deep-sea into what is now Figure S3 (previously Fig. S2). CIE C is off trend, but both CIE D and B plot in the trend of I1, I2, H2 and ETM2. We favor CIE D as correlative to hyperthermal U because it is the only CIE in our record with a magnitude larger than 2 standard deviation.*

[Figure]

The section 5.4 Preservation potential of hyperthermals in continental sections is informing but a bit out of context. It is clear to the community that higher sedimentation rates allow a more detail insight. This section, if kept in the manuscript, also needs to discuss that sedimentation is not uniform (steady) in terrestrial records but highly dynamic (50m away from the section things will look very different, see Bighorn Basin Project results where outcrops studies and drill cores allow a 3D view).

*We agree with referee #1's comment that partially echoes similar comments from referee #2.*

*We will reorganize chapter 5.4 as follow:*

*Line 398:*
*"5.4 **Possible implication for the** preservation potential of hyperthermals in continental deposits"*

*Lines 399 to 406 were removed*

*Lines 406:*
*"Major events such as the PETM event have proven to be detectable in both marine and continental environments (e.g.; Abels et al., 2016; Koch et al., 1992), but the signal and preservation potential of smaller scale climatic events (e.g. hyperthermal events L to W in Lauretano et al., 2016), **may be more difficult to detect** (Foreman and Straub, 2017**) because of the inherent highly dynamic nature of sedimentation in fluvial deposits**. **To address this issue in the present case study**, we calculated the compensation time scale (Tc) **of the Castissent Fm**."*

*Lines 419 to 424 were removed*

*Line 424:*
*"Using an average sedimentation rate of 0.17 **mm/yr and an average channel depth of 3.75m**, we obtained a mean Tc of 22,000 yrs**, which means that** hyperthermal events of 40 kyrs **duration (time-scale of hyperthermal U and preceding CIE) have the potential to** be recorded **despite fluvial system dynamics**."*

*Line 427:*
*"**Our estimate of preservation potential assumes steady sedimentation rates throughout the section. But, sedimentation in terrestrial records is not uniform (steady) but rather highly variable, resulting in spatial and temporal changes in facies and deposition rates ranging from < 0.1 to 1-2 mm/yr (Bowen et al., 2015; Kraus et al., 2015; Marriott and Wright, 1993). However, mean accumulation rates give a reasonable estimate approximating more realistic (i.e., variable) sedimentation rates as observed in the Bighorn Basin (Bowen et al., 2015).***
*Additionally, we analyse the vertical movement of the nearby structures to evaluate their potential influence on disrupting deposition at Chiriveta during Castissent times. The Chiriveta section was deposited near or at the axis of the Tremp-Graus basin (Nijman, 1998), which is bounded by the Bóixols thrust in the north and the Montsec thrust in the south (Marzo et al., 1988). The Tremp-Graus basin is transported as a piggy-back basin on the*

*Montsec thrust emerging at the time approximatively 4 km south of the studied section (Nijman, 1998). In the basin axis, subsidence is the highest with rates of 0.1 to 0.29 mm/yr (this study and Marzo et al., (1988)). Taking into account a vertical movement rate of the Montsec thrust of 0.03 to 0.1 mm/yr during the Castissent time-interval (based on a horizontal displacement of 7 km, a period of activity lasting 26 Ma and a thrust dip between 6° and 20° (Clevis et al., 2004; Farrell et al., 1987; Nijman, 1998; Whitchurch et al., 2011), we estimate that the vertical displacement is no more than equal to sedimentation rates in the basin axis. This is consistent with the general absence of growth strata in the basin axis, although growth strata can indeed be observed closer to the Montsec (Nijman, 1998).*
*The rates of accumulation, distance to the main structures, and characteristic compensation time scale, together suggest that hyperthermal events of ca. 40 kys duration can be recorded in the Castissent Fm. These results confirm that, despite its highly dynamic nature, fluvial sedimentation may contain valuable record of high-frequency events, even in active tectonic contexts."*

*Lines 448 to 458 were removed*

Abstract Line 23 – Hyperthermal cannot be "potential analogues, in the geological record, to the ongoing anthropogenic modification of global climate". Background conditions 50+ million years ago were much different. But the events can help to test the assumptions made by climate models and revise them for a better understanding of the climate system dynamics.

*Thanks. Comparing the geological past with the current situation is often done, but we agree that the analogy has limitations and that the genuine value of investigating hyperthermals must be more clearly exposed. Inspire by the referee's wording of it, we thus propose to change this sentence in the abstract to:*
*Line 24: "**Documenting** how the Earth system responded to rapid **climatic shifts during hyperthermals provides** fundamental **information to constrain climatic models.**"*

Line 44-45: remove "towards icehouse conditions eventually reached later in the Cenozoic"

*Removed.*

Line 48: "Turner et al., 2014" change to Kirtland-Turner et al. 2014 in the entire text

*Done*

Line 51: "e.g., Westerhold et al., 2018"; add Lourens et al. 2005, Sexton et al. 2011, Kirtland-Turner et al. 2014, Lauretano et al. 2015, 2016. They published the records used.

*Modified*

Line 52: "Early Thermal Maximum (ETM) 2, H2, I1, I2, and ETM3/X events"– correct to Eocene Thermal Maximum. Change the wording of the sentence to clarify the nomenclature (ETM is only for 1, 2, 3; H1 H2 etc. are from Cramer et al. 2003 revised by Lauretano: : :, Sexton et al. 2011 suggested the relative to magnetochron scheme, see Westerhold et al.). In Figure 1 the text refers to this, please streamline the manuscript text accordingly.

*The sentence was modified accordingly: "**Eocene** Thermal Maximum (ETM) 2 **and 3**, H2, I1 and I2 events.*
*Figure 1 was modified with the naming schemes of Cramer et al., (2003), Lauretano et al., (2016) and Westerhold et al., (2017).*
*The last sentence of chapter 2 was modified to:*

*Line 140: "…this period was marked by 4 hyperthermals labelled S**/C22rH3**, T**/C22rH4**, U**/C22rH5** and V**/C22nH1** (**Cramer et al., 2003**; Lauretano et al., 2016; Westerhold et al., 2017)."*

Line 54: "In the stratigraphic record, these events are primarily characterized by important negative carbon isotope excursions (NCIEs)" – rephrase!, they are characterized by a paired negative excursion in carbon and oxygen isotope data; do not use NCIE throughout as it confuses with the commonly used CIE (Carbon Isotope Excursion) abbreviation for the e.g. PETM.

*The sentence will be rephrased as suggested and all NCIE in the text will be modified to "CIE".*

Line 150: please specify the target material of the X-ray tubes (Mo, Rh?).

*The target material of the X-ray tubes is Cu. This information was added in chapter 3.3 Majors and trace element composition.*
*Line 179: "…using a PANalytical PW2400 XRF spectrometer **with copper tube (Cu)** at the University of Lausanne…"*

Line 180: provide tie-points to ODP 1263 as a table in the supplement and add the age to your data tables of isotope as well as XRF results.

*Tie-points have been added in the new Table S3. Age have been added in the Table S2*

One important thing would be to show images of the soil nodules from the Chiriveta record in the supplement.

*Images of the soil nodules have been added as well as an image of the iron oxides nodules mentioned at line 374 (Figure S1).*

*Several typographical corrections, sentence reformulations and minor precisions have as well been implemented in this second version of the manuscript. Below are listed the majors ones.*

*Line 250:*

*A sub-chapter **5.1.1 Identifying the CIE** was added.*

*Line 251:*

*"In continental successions, the carbon isotope composition of pedogenic carbonate nodules—which consists of calcareous concretions between 1 mm and 4 cm diameter formed in situ in the floodplain—**have been shown to be sensitive to environmental conditions during their formation (e.g., Millière et al., 2011a, 2011b), and are therefore a promising tool to track how environments respond to carbon cycle perturbation**  The carbon isotope composition of the soil carbonate nodules depend on the δ13C value of the **atmospheric CO2 and** soil CO2, which in turn is a function of the δ13C of the atmospheric CO2 , **and** the overlying plants, as well as **the** soil respiration **flux and the partial pressure of atmospheric CO2** (Abels et al., 2012; Bowen et al., 2004; Caves et al., 2016; Cerling, 1984). "*

*Line 267:*

*"…nodules, **which** is consistent with **a large compilation of data from eastern Eurasia (Caves Rugenstein and Chamberlain, 2018)**"*

*Line 302:*

*"…varies between 0.1-0.29 mm/y, **consistent with sedimentation rates reported for other Eocene floodplain successions (Kraus and Aslan, 1993)**."*

*Line 307:*

*A sub-chapter **5.1.2 Mechanisms causing the CIE** was added.*

*Line 319:*

*"**A release of 500 to 1500 Gt of carbon in the form of methane would imply a marine CIE of 0.8 to 2.3‰ or 0.3 to 0.9‰ if the carbon origin is dissolved organic carbon (DOC) (Sexton et al., 2011). The latter seems more plausible regarding the observed amplitude of ~1‰ measured in the marine record for hyperthermal U (Westerhold et al., 2017) and the supposed origin linked to the oxygenation of deep-marine DOC of post-PETM hyperthermals (Sexton et al., 2011). A global shift of -1‰ in δ13C can however not fully explain the 3‰ shift in δ13C observed in this study.** "*

---

## Author Comment (AC2) · 7 Nov 2019

*In this response, the original comments are in black and responses by the authors to the reviews are written in blue. Changes in the manuscript are written in red.*

*We thank Referee #2 for their careful review that has allowed us to clarify several important points and improve the manuscript.*

1. The title of the manuscript (Alluvial record of an early Eocene hyperthermal, Castissent Formation, Pyrenees, Spain) is misleading: When I first read it I though the authors meant that the Castissent Fm had resulted from the effects of an hyperthermal – which would be impossible. I therefore suggest something like: Alluvial record of an early Eocene hyperthermal within the Castissent Formation, Pyrenees, Spain.

*We agree with the reviewer. Indeed the Castissent fm cannot be associated with an hyperthermal in its entirety because its duration of ca. 500 ka is much longer than any of the known hyperthermals of that period (between 50.5 and 49.7 Ma, i.e. S, T, U, V.). Therefore, we agree that the title could be somewhat misleading in that respect and we will follow the suggestion of the reviewer to modify the manuscript title:*

*"Alluvial record of an early Eocene hyperthermal **within the** Castissent Formation, Pyrenees, Spain"*

2. In lines 36–40 they state that "The results show that even relatively small-scale hyperthermals compared with their prominent counterparts, such as PETM, ETM2 and 3, have left a recognizable trace in the stratigraphic record. . .". The environmental effects of the above-mentioned hyperthermals in terrestrial setting, especially those of the PETM, are very prominent (e.g., Foreman et al, 2012), a fact that made their record easily recognizable in the field. For the alleged U event (NCIE D) the information provided does not justify the above assertion. Thus, in Fig. 7 the event seems to be represented by a comparatively thin interval (<1 m), whereas the previous NCIE C, which is of smaller magnitude, is recorded by a ∼3 m thick interval. A better documentation of the sedimentological features of the alleged U event is needed, as it is the focus of the study. It would also help if such (distinctive?) features could be observed in other section(s).

*This sentence in the abstract can indeed be misleading with respect to the nature of our results. In our text, "recognizable **trace**" simply referred to the geochemical signature of the event, and not to its sedimentological "recognizable trace" on the field. In the present paper, our main point is to suggest that the U event is marked in the succession by its geochemical signature, which is not trivial given the often-assumed low preservation potential of fluvial*

*deposits. Therefore, to reflect this comment of reviewer #2, we propose to change the sentence to:*

*Line 39: "The results show that even relatively small-scale hyperthermals compared with their prominent counterparts, such as PETM, ETM2 and 3, have **can leave** a recognizable **signature** in the **terrestrial** stratigraphic record. . ."*

*About the point made by the reviewer on the correlative character of this event in other sections: we agree that it would help to distinguish the same signature in coeval successions in order to support the regional (supra-regional) nature of the driver behind the excursion. However, unfortunately, we have not yet assembled sufficient data to assess this into other sections laterally. Distinctive red intervals within the Castissent Fm do occur in the Isábena valley (Marzo et al., 1988) but we are not yet in a position to say if those correlate with the negative CIE observed in the Chiriveta section, primarily because of a lack of temporal constraints at this resolution. We currently collaborate on developing a magnetostratigraphic frame for these deposits, but the results of this endeavour are beyond the scope of the present study.*

3. Constraint on the age of the Castissent Fm is somewhat vague. It is not based on data from the Chiriveta section itself, but on bio- and magnetostratigraphic studies of previous authors (Kapellos and Schaub, 1973; Bentham and Burbank, 1996), carried out in the Campo section, 40km westward. Based on them the authors indicate that the Castissent Fm occurs within the D. lodoensis nannoplankton zone (= NP 13), with the base and top of the nannozone being respectively situated at ca 200 m below the base of the Castissent, and at ca 100 m above its top. My doubts about the reliability of the Kapellos and Schaub zonation (1973) partly stem from the fact that shallow marine facies such as those of the Campo section are not favorable for the preservation of nannofossils, and that therefore are not entirely reliable. The NP9/NP10 boundary provides a proof of this, for K & Sch073 did situate it ABOVE the so-called Alvelina limestone, while Orue-Etxebarria et al. (2001; Marine Micropaleontology 41, 45-71) proved that it occurred BELOW such unit, a finding that permitted to correctly place the PETM interval in the Campo section (see Fig. 1 of this Comment). More to the point, as shown also in Fig 1, the location of the top NP 13 zone is somewhat ambiguous: K & Sch073 state in their text that the NP 13 zone spans from km 58.6 (base) to km 56 (top), whereas in their columnar section the top of the zone is placed at sample 32. Such uncertainty raises doubts about the magnetostratigraphic calibration of Bentham and Burbank (1996), likely based on the K & Sch073. Indeed, in Fig. 3 of the manuscript the NP13/14 boundary is placed within the C22n magnetozone, whereas in Fig. 1 (from Westerhold et al, 2017) is located within C23r.

*Defining with confidence the age of the fluvial Castissent formation is indeed a challenge. The question of the age model of the Castissent is also raised by reviewer #3.*
*The only time-constraint available for the study location itself is that of remains found in the Chiriveta area and belonging to European Mammals zone MP10 (Badiola et al., 2009). In order to better constrain independently the Chiriveta section itself, we studied the option of U/Pb*

*dating on pedogenetic carbonate nodules; however the analysis error of recent work (Methner et al., 2016) lays between 0.8-1.4 Ma, which is critical on our 0.8 Ma interval.*

*We therefore rely, as mention by Reviewer #2, on bio- and magnetostratigraphic studies of previous authors carried out in the Campo section, 40km westward such as Kapellos and Schaub, (1973) and Bentham and Burbank (1996) as well as Marzo et al. (1988), Tosquella (1995) and Payros et al. (2009).*

*Reviewer #2 points out the difference of the position of the NP13/NP14 regarding magnetozone C22n in our figures 1 and 3. In figure 3 of our manuscript, the limit between NP13 and NP14 is placed within Chron C22n based on the available data in this section (i.e., biostratigrafic data from Kapellos and Schaub 1973 (K&Sch73) and the magnetozones of Bentham and Burbank 1996 (B&B96)). As mentioned by reviewer #2, this is not in line with Fig. 1 which show the location of NP13/NP14 at the base of C21r as reported in GTS 2012. This difference had already been observed by Payros et al., (2009) (their figure 9). Although the position of the above-mentioned limit is disputable and based on available data, it doesn't affect the upper part of our age model because, to constrain the Castissent Formation, we use the limit between C22r and C22n which is below NP13/NP14 in both models and is at 49.695±0.043 Ma according to the recent astrochronologic age models of Westerhold et al. (2017).*

*Reviewer #2 further raises a concern about the validity of B&B96 magnetostratigraphy in the Campo section because "the calibration of Bentham and Burbank (1996), [is] likely based on the K & Sch073". We note however that the work of K&Sch73 is not cited in the paper of B&B96, so it is not clear to us to what extent the magnetostratigraphic age model of B&B96 should be taken with caution. For instance, B&B96 place the base of the Campo section, and the so-called Alveolina limestone, in C24r. According to the updated biostratigraphy of this section cited by reviewer #2 (Orue-Etxebarria et al., 2001), the Alveolina limestone is deposited during NP10, which is still in C24r. We therefore consider that the magnetozone interpretation of B&B96 in this section may still be valid, although we are open to more data and constraints if available.*

*However, for the lower time-constraint of the Castissent formation used in this study (i.e., the limit between NP12 and NP13 at ca. 50.5), we rely on the biostratigraphy work of K&Sch73, Tosquella (1995) and Payros et al. (2009).*

*In figure S2, we however investigate several correlation options encompassing different climatic scenarios in the time-interval inferred by the bio- and magnetostratigraphic studies of previous authors in order to suggest the most plausible correlation between global and local isotopic record.*

*Considering these observations, we will modify the paragraph as follow and added a supplementary Table (S1) regarding slope estimation:*

*Line 115: "**In the Chiriveta location, stratigraphic constraints are limited to the identification of European Mammals zone MP10 (Badiola et al., 2009), which gives an age range of between 50.73 to 47.4 Ma (GTS2012). This age span is refined by bio- and magnetostratiphic data from the Castissent Fm. outcrops of the Campo location, about 40 km further west**

*(Bentham and Burbank, 1996; Kapellos and Schaub, 1973; Payros et al., 2009; Tosquella, 1995; Tosquella et al., 1998) (fig. 3).* *Because of its outcropping extent, the Castissent Fm. has been mapped from west to east across these sections (Chanvry et al., 2018; Nijman, 1998; Nijman and Nio, 1975; Poyatos-Moré, 2014)*. **The low slope of the Castissent Fm. (ca. 2.3 × 10$^{-4}$ m/m, see supplementary Table S1) indicate an elevation drop of ca. 1 m between the Chiriveta section and the Campo section. Given an average flow depths of 3.75 m in the Castissent channels based on measurement in the Chiriveta and La Roca sections, we thus assume no significant time-lag of deposition between both sections.**
*[…]*
*Line 139:* **Considering the data available and their resolution, we suggest a depositional age span** *between* **50.5** *and* **49.7** *Ma for the Castissent Fm (reported in green on Fig. 1)."*

4. The completeness of the studied section is debatable In the first paragraphs of chapter 5.4 ("Preservation potential of hyperthermals in continental sections"), the authors acknowledge that alluvial-fluvial stratigraphic records are considered incomplete by many authors (e.g., Shanley and McCabe, 1994; Wright and Marriott, 1993; Turner et al., 2015; Barrell, 1917; Sadler, 1981). In the present case, Marzo et al. (1988) concluded that "The sedimentation of the Castissent Formation was structurally controlled by an interplay of vertical basement movement due to thrust stacking in the hinterland and surficial thrust displacement to the foreland resulting in alternating southward and northward shift of the fluvial system". The Chiriveta section is close to the foreland thrust (Montsec thrust) and, in such dynamic scenario, it is doubtful that it would have accumulated a (near) continuous succession. But, even if that were the case, it seems rather improbable that the section would be complete enough to have recorded ALL the minor NCIES detected in the ODP 1263 site, as shown in Fig. 7.

*Referee #2 refers, rightly, to a long-standing debate in Earth sciences: the completeness of the stratigraphic record. This debate is beyond the reach of our study, but we want to contribute because we believe that our findings, if correct, may suggest that alluvial records may be less punctuated than sometimes considered. There are two aspects here in the reviewer comment that we wish to respond to: 1) the inherent incompleteness of fluvial stratigraphy, and 2) incompleteness due to structural deformation in an active basin.*

1. *To assess whether it was plausible or not that the Castissent Fm. recorded hyperthermal events, in the second part of chapter 5.4, we calculated the compensation time scale (Tc) of the formation, which represents an estimate of the autogenic time-scale of the fluvial system linked to avulsion processes (Wang et al., 2011) . The Tc obtained for this study is of 22,000 yrs, i.e. twice as short as the inferred duration of the hyperthermal U and smaller CIEs preceding it, which have typical durations of ca. 40,000 yrs. According to this perspective, it is therefore not unrealistic that such "events" are recorded in the Castissent fm. We also develop this point in our response to referee #1.*

2. *Based on paleogeographical reconstructions, the Castissent Formation, at the Chiriveta section, is deposited near or at the axis of the Tremp-Graus basin, transported on the back of the Montsec thrust and approximately 4km away from the thrust emergence (Nijman, 1998). In this area, subsidence is the highest, with rates of between 0.1 and 0.29 mm/yr (this study and Marzo et al., 1988). This represents between 50 and 150m of accumulation during the Castissent time interval (0.8 Ma).*

*Based on an inferred total horizontal displacement of the Montsec of 7 km (Whitchurch et al., 2011, Farrell et al., 1987), a period of activity lasting 26 Ma (Whitchurch et al., 2011) and a thrust dip between 6° and 20° (Clevis et al., 2004), we estimate a vertical movement of between 25 and 90 m during the Castissent time-interval. This vertical displacement is thus no more than equal to sedimentation rate in the basin axis. This is consistent with the general absence of growth strata in the basin axis, although growth strata can indeed be observed closer to the Montsec.*

*In conclusion, the rates of accumulation, distance to the main structures, and characteristic compensation time scale together suggest that hyperthermal events of ca 40ky duration can be plausibly recorded in the Castissent Fm, despite its situation in a tectonically active fluvial basin.*

*We agree with referee #2 comment, which is complementary to some of referee #1's comments and we will therefore develop and reorganize chapter 5.4 consequently.*

*Line 398:*
*"5.4 **Possible implication for the** preservation potential of hyperthermals in continental deposits"*

*Lines 399 to 406 were removed*

*Lines 406:*
*"Major events such as the PETM event have proven to be detectable in both marine and continental environments (e.g.; Abels et al., 2016; Koch et al., 1992), but the signal and preservation potential of smaller scale climatic events (e.g. hyperthermal events L to W in Lauretano et al., 2016), **may be more difficult to detect** (Foreman and Straub, 2017**) because of the inherent highly dynamic nature of sedimentation in fluvial deposits**. **To address this issue in the present case study**, we calculated the compensation time scale (Tc) **of the Castissent Fm**."*

*Lines 419 to 424 were removed*

*Line 424:*
*"Using an average sedimentation rate of 0.17 **mm/yr and an average channel depth of 3.75m**, we obtained a mean Tc of 22,000 yrs**, which means that** hyperthermal events of 40 kyrs **duration (time-scale of hyperthermal U and preceding CIE) have the potential to** be recorded **despite fluvial system dynamics**."*

*Line 427:*
*"**Our estimate of preservation potential assumes steady sedimentation rates throughout the section. But, sedimentation in terrestrial records is not uniform (steady) but rather highly variable, resulting in spatial and temporal changes in facies and deposition rates ranging from < 0.1 to 1-2 mm/yr (Bowen et al., 2015; Kraus et al., 2015; Marriott and Wright, 1993). However, mean accumulation rates give a reasonable estimate approximating more realistic (i.e., variable) sedimentation rates as observed in the Bighorn Basin (Bowen et al., 2015).***

*Additionally, we analyse the vertical movement of the nearby structures to evaluate their potential influence on disrupting deposition at Chiriveta during Castissent times. The Chiriveta section was deposited near or at the axis of the Tremp-Graus basin (Nijman, 1998), which is bounded by the Bóixols thrust in the north and the Montsec thrust in the south (Marzo et al., 1988). The Tremp-Graus basin is transported as a piggy-back basin on the Montsec thrust emerging at the time approximatively 4 km south of the studied section (Nijman, 1998). In the basin axis, subsidence is the highest with rates of 0.1 to 0.29 mm/yr (this study and Marzo et al., (1988)). Taking into account a vertical movement rate of the Montsec thrust of 0.03 to 0.1 mm/yr during the Castissent time-interval (based on a horizontal displacement of 7 km, a period of activity lasting 26 Ma and a thrust dip between 6° and 20° (Clevis et al., 2004; Farrell et al., 1987; Nijman, 1998; Whitchurch et al., 2011), we estimate that the vertical displacement is no more than equal to sedimentation rates in the basin axis. This is consistent with the general absence of growth strata in the basin axis, although growth strata can indeed be observed closer to the Montsec (Nijman, 1998).*
*The rates of accumulation, distance to the main structures, and characteristic compensation time scale, together suggest that hyperthermal events of ca. 40 kys duration can be recorded in the Castissent Fm. These results confirm that, despite its highly dynamic nature, fluvial sedimentation may contain valuable record of high-frequency events, even in active tectonic contexts."*

*Lines 448 to 458 were removed*

5. Section 4.1 of the manuscript ("Overview 0f the Castissent Fm at the Chiriveta section) seems to be misplaced. I suggest to remove it from the Results section and place it after the Chapter 2, Geological setting.

*We understand Reviewer #2's point of view as section 4.1 didn't specified enough that the section logged in this study is not based on a previous work. We would however prefer to keep this section at its current place because we think our description is an integral part our study. We will modify the title and introduction of section 4.1 to stress this point.*

*Line 203:*
*"4.1 **Sedimentology of the Castissent Formation at Chiriveta***
***We here describe the section logged and sampled in this work (Fig. 4). At Chiriveta**, the Castissent Fm. is a paleosol-rich succession, which shows greyish-yellow…"*

6. I have not had the time to check out all the references, but in a quick glance I can point out that some of them are incomplete:

   Hunger, T.: Climatic signals in the Paleocene fluvial formation of the Tremp-Graus Basin, Pyrenees, Spain. University of Geneva., 2018. Is that a Thesis? How many pages? It is published or unpublished?

*Completed.*
*It's a master thesis, published on the open archives of the University of Geneva. It is however not an open access document.*

*The correct reference reads:*

**Hunger, T.: Climatic signals in the Paleocene fluvial formation of the Tremp-Graus Basin, Pyrenees, Spain, MSc Thesis, University of Geneva, pp. 123. https://archive-ouverte.unige.ch/unige:124264, 2018**

Poyatos-Moré, M.: Physical Stratigraphy and Facies Analysis of the Castissent Tecto-Sedimentary Unit., 2014. Is that a Thesis? If so, from which University? How many pages? It is published or unpublished?

*Modified. It's a PhD thesis.*

*The correct reference reads:*

*Poyatos-Moré, M.: Physical Stratigraphy and Facies Analysis of the Castissent Tecto-Sedimentary Unit, PhD Thesis, Universidad Autónoma de Barcelona, pp. 284. https://ddd.uab.cat/record/127119, 2014*

The list of authors of the reference "Payros, A. and Tosquella, J.: Filling the North European Early/Middle Eocene (Ypresian/Lutetian) boundary gap: insights from the Pyrenean continental to deep-marine record, Palaeogeogr. Palaeoclimatol. Palaeoecol., 280, 313–332, doi:10.1016/j.palaeo.2009.06.018, 2009" is incomplete. Either include all the authors (Payros, A., Tosquella, J., Bernaola, G., Dinarès-Turell, J., Orue-Etxebarria, X., and Pujalte, V.,), or quote it as Payros, A., Tosquella, J, et al.

*Thanks. Modified*

7.  Some previous papers should be referenced. In lines 60-63 the manuscript states that "In coastal marine sections, Early Eocene hyperthermal events are generally associated with an enhanced flux of terrigenous material, interpreted as linked to accelerated hydrological cycle and higher seasonality (Bowen et al., 2004; Dunkley Jones et al., 2018; Nicolo et al., 2007; Payros et al., 2015; Slotnick et al., 2012): : :" To my knowledge, one of the first paper pointing out this fact was: Schmitz, B., Pujalte, V., Núñez-Betelu, K., 2001. Climate and sea-level perturbations during the Initial Eocene Thermal Maximum: evidence from siliciclastic units in the Basque Basin (Ermua, Zumaia and Trabakua Pass), northern Spain. Palaeogeogr. Palaeoclimatol. Palaeoecol. 165, 299–320

*This is correct. Thanks for noticing. We have added this reference to the manuscript.*

In lines 63-65 the manuscript states that "Several studies document a spatially heterogeneous hydrological climatic response during the PETM (Bolle and Adatte, 2001; Carmichael et al. 2017; Kraus and Riggins, 2007)". The paper by Giusberti, L., Boscolo Galazzo, F., Thomas, E., 2016. Variability in climate and productivity during the Paleocene–Eocene Thermal Maximum in the western Tethys (Forada section). Clim. Past 12, 213–240, should be acknowledged, as their compilation made evident such climatic variability.

*Thank you for pointing out this study. We added this reference to the manuscript.*

*It was as well added line 386:*

*"Such a climatic behaviour, was already described for the PETM, during the pre-onset excursion (Bowen et al., 2014)* **and in the core CIE of the PETM (Giusberti et al., 2016)***…"*

*Several typographical corrections, sentence reformulations and minor precisions have as well been implemented in this second version of the manuscript. Below are listed the majors ones.*

*Line 250:*
*A sub-chapter* **5.1.1 Identifying the CIE** *was added.*
*Line 251:*
*"In continental successions, the carbon isotope composition of pedogenic carbonate nodules—which consists of calcareous concretions between 1 mm and 4 cm diameter formed in situ in the floodplain—***have been shown to be sensitive to environmental conditions during their formation (e.g., Millière et al., 2011a, 2011b), and are therefore a promising tool to track how environments respond to carbon cycle perturbation**  *The carbon isotope composition of the soil carbonate nodules depend on the δ13C value of the* **atmospheric CO2 and** *soil CO2, which in turn is a function of the δ13C of the atmospheric CO2 ,***and** *the overlying plants, as well as* **the** *soil respiration* **flux and the partial pressure of atmospheric CO2** *(Abels et al., 2012; Bowen et al., 2004; Caves et al., 2016; Cerling, 1984). "*

*Line 267:*
*"…nodules,* **which** *is consistent with* **a large compilation of data from eastern Eurasia (Caves Rugenstein and Chamberlain, 2018)***"*

*Line 302:*
*"…varies between 0.1-0.29 mm/y,* **consistent with sedimentation rates reported for other Eocene floodplain successions (Kraus and Aslan, 1993)***."*

*Line 307:*
*A sub-chapter* **5.1.2 Mechanisms causing the CIE** *was added.*

*Line 319:*
**"A release of 500 to 1500 Gt of carbon in the form of methane would imply a marine CIE of 0.8 to 2.3‰ or 0.3 to 0.9‰ if the carbon origin is dissolved organic carbon (DOC) (Sexton et al., 2011). The latter seems more plausible regarding the observed amplitude of ~1‰ measured in the marine record for hyperthermal U (Westerhold et al., 2017) and the supposed origin linked to the oxygenation of deep-marine DOC of post-PETM hyperthermals (Sexton et al., 2011). A global shift of -1‰ in δ13C can however not fully explain the 3‰ shift in δ13C observed in this study.** *"*

---

## Author Comment (AC3) · 7 Nov 2019

**Response to "Review of "Alluvial record of an early Eocene hyperthermal, Castissent Formation, Pyrenees, Spain""**

Louis Honegger et al.

*In this response, the original comments are in black and responses by the authors to the reviews are written in blue. Changes in the manuscript are written in red.*

*We thank Referee #3 for their work and appreciation of our study and the valuable propositions of improvements suggested. We answer below point by point to each item of concern.*

1) I think this paper would benefit from a more detailed description of the model used to constrain the age of these deposits. Was the placement of the Castissent Formation within European Mammal Zone MP10 based on the same outcrops sampled here, and if not, what is the proximity of that site? Although the authors state that many of the well-dated sections within the Castissent Formation can be physically correlated to the current study area, how can the authors be confident that these are not time-transgressive deposits? Finally, I am skeptical that the age designation bracket of 50.534-0.025 and 49.695-0.043 Ma can be realistically applied to this unit. That extremely precise age range is based on orbital tuning of a marine record, correlated to a continental record, correlated to the current study area. I am not disputing the correlation, just that the precision of the marine record might not be retained through two iterations of lithostratigraphic correlation.

*European Mammal Zone MP10 was found next to the Chiriveta section which is the only direct dating available. A first-hand estimation of the slope of the Castissent Fm. based on grain-size give a slope of $1*10^{-4}$. This would imply a difference of elevation of 1 m between the Campo section (40 km away from the study area, where most of the dating is done) and the Chiriveta section. Considering a mean sedimentation rate of 0.17 m/kyr at the Chiriveta section (this study), it represents a potential lag of ca. 6 kyr between both sections. Therefore, we can make the assumption that the time between upstream or downstream deposition is short enough to not alter significantly the correlation between both sections. Moreover, the correlation of the under- and overlying formations (Castigaleu and Perrarua respectively) adds an additional constraint. A short diachronicity cannot be ruled-out but synchronism on the scale of these two outcrops is a reasonable hypothesis.*

*We however agree with Referee #3 comments that the extreme age precision from orbital tuning cannot be applied as such for our section. We acknowledge the dating precision of the author in the previous sentence, but ages are rounded up for the Castissent extension. We will change the manuscript accordingly:*

*Line 115: "**In the Chiriveta location, stratigraphic constraints are limited to the identification of European Mammals zone MP10 (Badiola et al., 2009), which gives an age range of***

*between 50.73 to 47.4 Ma (GTS2012). This age span is refined by bio- and magnetostratiphic data from the Castissent Fm. outcrops of the Campo location, about 40 km further west (Bentham and Burbank, 1996; Kapellos and Schaub, 1973; Payros et al., 2009; Tosquella, 1995; Tosquella et al., 1998) (fig. 3).* *Because of its outcropping extent, the Castissent Fm. has been mapped from west to east across these sections (Chanvry et al., 2018; Nijman, 1998; Nijman and Nio, 1975; Poyatos-Moré, 2014)*. *The low slope of the Castissent Fm. (ca. 2.3 × 10$^{-4}$ m/m, see supplementary Table S1) indicate an elevation drop of ca. 1 m between the Chiriveta section and the Campo section. Given an average flow depths of 3.75 m in the Castissent channels based on measurement in the Chiriveta and La Roca sections, we thus assume no significant time-lag of deposition between both sections.*

*[…]*

*Line 138:* *Considering the data available and their resolution, we suggest a depositional age span* *between* *50.5* *and* *49.7* *Ma for the Castissent Fm (reported in green on Fig. 1)."*

2) The authors note that unlike most marine hyperthermal records, the oxygen and carbon isotopic records are not coupled in the Castissent Formation (the oxygen does not reflect hyperthermal events, whereas the carbon does). Why might this be? Is there evidence of isotopic resetting of the O system (petrographic or other)? How deeply have these rocks been buried? This seems to suggest that even in well-preserved systems, oxygen isotopic records should be used and viewed with caution.

*Frequently, oxygen and carbon isotopes are not coupled during hyperthermal events in continental record as already observed by Schmitz and Pujalte, (2003), Bowen et al., (2001) for the PETM. Similarly, small changes in δD are also frequently observed in mid-latitude CIE section (Smith et al. 2007; Tipple et al. 2011). Though the precise mechanisms that produce stable δ$^{18}$O/δD are still debated, mid-latitude precipitation δ$^{18}$O appears to be relatively insensitive to changes in atmospheric pCO$_2$ and warming, particularly in greenhouse climates (Winnick et al. 2015). Further, the stable δ$^{18}$O value (around -5.5‰) throughout the Chiriveta section is likely additionally stabilized by its position close to the coast, which will buffer coastel precipitation δ$^{18}$O values relative continental interiors (Kukla et al 2019). This coastal influence is clearly seen in Figure 6 where oxygen isotopes values of the Bighorn Basin have a more continental (i.e., more negative) δ$^{18}$O signature then that in the Pyrenees, which is in line with the paleogeography of both basins at the time. This consistency between our results and the ones from the Bighorn Basin suggests a relative good preservation of the oxygen isotopic signal.*

*However, because sediments from this section might have been buried between 2-3 km and that oxygen isotopes are more prone to diagenesis than carbon oxygen and might therefore not preserve a primary signal, we use them with caution.*

*The manuscript was modified accordingly:*

*Line 267:*
*"Pre-PETM δ$^{18}$O values from carbonate nodules from the same area (–4.5 ± 0.4 ‰) (Hunger, 2018) show similar values than the Chiriveta section's measurements (–6.0 ± 0.4 ‰).* *Oxygen and carbon isotopes are not coupled during hyperthermal events in continental record as already observed by Schmitz and Pujalte, (2003), Bowen et al., (2001)* *for the PETM isotopic excursion.* *Though the precise mechanisms that produce stable δ$^{18}$O during CIE are still*

*debated, mid-latitude precipitation δ¹⁸O appears to be relatively insensitive to changes in atmospheric pCO₂ and warming, particularly in greenhouse climates (Winnick et al. 2015). In contrast, the stable δ¹⁸O value of soil carbonates from the Pyrenean foreland basin (excluding the PETM) (−5.3 ± 0.9 ‰) is likely additionally stabilized by its position close to the coast (Cerling, 1984, Kukla et al 2019) compared for example to those of the Bighorn Basin (−9.0 ± 0.6 ‰). This is in line with a more continental paleogeographical position of the Bighorn Basin compared to the Tremp-Graus Basin at the time (Seeland, 1998)."*

*Several typographical corrections, sentence reformulations and minor precisions have as well been implemented in this second version of the manuscript. Below are listed the majors ones.*

*Line 250:*

*A sub-chapter **5.1.1 Identifying the CIE** was added.*

*Line 251:*

*"In continental successions, the carbon isotope composition of pedogenic carbonate nodules—which consists of calcareous concretions between 1 mm and 4 cm diameter formed in situ in the floodplain—**have been shown to be sensitive to environmental conditions during their formation (e.g., Millière et al., 2011a, 2011b), and are therefore a promising tool to track how environments respond to carbon cycle perturbation**  The carbon isotope composition of the soil carbonate nodules depend on the δ13C value of the **atmospheric CO2 and** soil CO2, which in turn is a function of the δ13C of the atmospheric CO2 ,**and** the overlying plants, as well as **the** soil respiration **flux and the partial pressure of atmospheric CO2** (Abels et al., 2012; Bowen et al., 2004; Caves et al., 2016; Cerling, 1984). "*

*Line 267:*

*"…nodules, **which** is consistent with **a large compilation of data from eastern Eurasia (Caves Rugenstein and Chamberlain, 2018)**"*

*Line 302:*

*"…varies between 0.1-0.29 mm/y, **consistent with sedimentation rates reported for other Eocene floodplain successions (Kraus and Aslan, 1993)**."*

*Line 307:*

*A sub-chapter **5.1.2 Mechanisms causing the CIE** was added.*

*"A release of 500 to 1500 Gt of carbon in the form of methane would imply a marine CIE of 0.8 to 2.3‰ or 0.3 to 0.9‰ if the carbon origin is dissolved organic carbon (DOC) (Sexton et al., 2011). The latter seems more plausible regarding the observed amplitude of ~1‰ measured in the marine record for hyperthermal U (Westerhold et al., 2017) and the supposed origin linked to the oxygenation of deep-marine DOC of post-PETM hyperthermals (Sexton et al., 2011). A global shift of -1‰ in δ13C can however not fully explain the 3‰ shift in δ13C observed in this study. "*